# Chromosome-scale assemblies of the male and female *Populus euphratica* genomes reveal the molecular basis of sex determination and sexual dimorphism

Shanhe Zhang [1,5], Zhihua Wu [2,5], De Ma [3,5], Juntuan Zhai [1], Xiaoli Han [1], Zhenbo Jiang[1], Shuo Liu [4], Jingdong Xu[4], Peipei Jiao [1✉] & Zhijun Li [1✉]

Reference-quality genomes of both sexes are essential for studying sex determination and sex-chromosome evolution, as their gene contents and expression profiles differ. Here, we present independent chromosome-level genome assemblies for the female (XX) and male (XY) genomes of desert poplar (*Populus euphratica*), resolving a 22.7-Mb X and 24.8-Mb Y chromosome. We also identified a relatively complete 761-kb sex-linked region (SLR) in the peritelomeric region on chromosome 14 (Y). Within the SLR, recombination around the partial repeats for the feminizing factor *ARR17* (*ARABIDOPSIS RESPONSE REGULATOR 17*) was potentially suppressed by flanking palindromic arms and the dense accumulation of retro-transposons. The inverted small segments S1 and S2 of *ARR17* exhibited relaxed selective pressure and triggered sex determination by generating 24-nt small interfering RNAs that induce male-specific hyper-methylation at the promoter of the autosomal targeted *ARR17*. We also detected two male-specific fusion genes encoding proteins with NB-ARC domains at the breakpoint region of an inversion in the SLR that may be responsible for the observed sexual dimorphism in immune responses. Our results show that the SLR appears to follow proposed evolutionary dynamics for sex chromosomes and advance our understanding of sex determination and the evolution of sex chromosomes in *Populus*.

[1] College of Life Sciences and Technology, Tarim University/Key Laboratory of Protection and Utilization of Biological Resources in Tarim Basin, Xinjiang Production & Construction Corps/Research Center of Populus Euphratica, Aral 843300, China. [2] College of Chemistry and Life Sciences, Zhejiang Normal University, Jinhua 321004, China. [3] Novogene Bioinformatics Institute, Beijing 100083, China. [4] Hubei Provincial Key Laboratory for Protection and Application of Special Plant Germplasm in Wuling Area of China, College of Life Sciences, South-Central Minzu University, Wuhan 430074, China. [5]These authors contributed equally: Shanhe Zhang, Zhihua Wu, De Ma. ✉email: jiaopeipei2000@126.com; lizhijun0202@126.com

How sex determination evolved to produce hermaphroditic (with both sexes in the same flower), monecious (one sex per flower, both sexes on the same plant), and diecious (one sex per flower, each sex on separate plants) species is a fascinating and complex mystery. Diecious plants ensure outcrossing and have multiple evolutionary origins. Sex in diecious species is usually determined by a pair of sex chromosomes represented by either male (XY system) or female (ZW system) heterogamety[1]. The canonical two-factor model for the emergency of dioecy assumes that two sex-determining genes will become linked on one chromosome, one affecting female function and one male function[2,3], which is supported by experimental data of kiwifruit (*Actinidia deliciosa*)[4] and asparagus (*Asparagus officinalis*)[5]. However, the one-factor model proposing a single master regulator of sex switch gene *ARR17* is also raised and verified by CRISPR-Cas9-induced mutation in *Populus tremula*[6]. Additionally, there was proposed a unified sex determination model: similar genes and pathways may determine the sex of several diecious species, regardless of one-factor or two-factor model[7].

After the emergence of the sex-determination gene (or genes), a pivotal event in sex-chromosome evolution is the suppression of recombination flanking this gene (or these genes)[1,2,8,9]. The cessation of recombination is associated with diverse factors such as chromosomal rearrangements, especially inversions in papaya (*Carica papaya*)[10], humans (*Homo sapiens*)[11], and white-throated sparrow (*Zonotrichia albicollis*)[12,13], as well as expansions of repetitive sequences in some songbirds[14] and other unknown factors[15]. The resulting palindromes in the sex-determination region along the Y or W chromosome reported in great apes[16], purple willow (*Salix purpurea*)[17], and black cottonwood (*Populus trichocarpa*)[18] allow arm-to-arm gene conversion[19], which protects the non-recombining male-specific regions from deleterious mutations[17,20]. On the other hand, if extended regions that suppress recombination of a sex-determining gene (or genes) do not evolve first, the size of sex-determining regions may decouple with their age, and sex chromosomes may remain homomorphic[3].

Member species of the genus *Populus* (Salicaceae family) are exclusively dioecy with various sexual systems, sex chromosomes, and sex-determination regions (SDRs)[6,7,21–23]. Among them, *P. alba* (probably also *P. adenopoda* and *P. qiongdaoensis*) has a ZW system[6,21,23], while the rest species in this genus bear an XY system[6,7,21–23]. This genus, therefore, provides an ideal model to explore the mechanism(s) of sex determination, sex-chromosome evolution, and the turnover of sexual systems. So far, several female and male genomes had been available for some species in the genus, such as white poplar (*P. alba*)[6], Eastern cottonwood (*P. deltoides*)[22], desert poplar (*P. euphratica*)[21,24], European aspen (*P. tremula*)[6] and black cottonwood (*P. trichocarpa*)[18,25]. The analysis of these genomes revealed that the sex of *P. tremula* trees is determined by the expression of the poplar ortholog to *ARR17* (*ARABIDOPSIS RESPONSE REGULATOR 17*) within the pseudo-autosomal region on chromosome 19 (Chr. 19, Y), which is targeted by small RNAs arising from partial duplicates of *ARR17* in the SDR on Chr. 19[6]. In contrast, sex in *P. deltoides* trees is controlled by two Y-specific genes, *MSL* (male-specific long non-coding RNA) and *FERR-R* (the repressor of female-specifically expressed *RESPONSE REGULATOR*)[22]. Partial *RR* (*RESPONSE REGULATOR*) segments are also present in the SDR of Chr. 14 in *P. euphratica* with a male heterogametic system[21]. Thus, the sex-determination mechanism and the sex-chromosome evolution in *Populus* are complex. The independent de novo assembly of female and male genomes and the phasing of sex chromosome are very important for the elucidation of these questions. However, thus far, no study has independently assembled both the female and male genomes. In particular, the structural differences

and evolution of sex chromosomes, and the elements within the SDR, are unknown in *P. euphratica*.

Sexual dimorphism refers to the complement of characteristics differentiating two sexes of the same species (such as diecious plants). In animals, sexual dimorphism encompasses differences in external appearance as well as internal organs and biological functions, such as the immune system, which plays an important role in infectious disease susceptibility[26–30]. Male and female plants may be exposed to different selection pressures for resource acquisition that may, in turn, lead to the evolution of sexual dimorphism and dimorphic adaptation to ecological niches[31–33]. Male and female individuals of many diecious species also exhibit differences in plant phenology[34], photosynthetic performance[35], water use efficiency[36], and herbivory tolerance[37]. Several Salicaceae species, such as Cathay poplar (*P. cathayana*)[38–40], quaking aspen (*P. tremuloides*)[41], and Arctic willow (*S. arctica*)[42], show sexual dimorphism in their adaptive responses to various environmental stress conditions. For example, male *P. tremuloides* trees have a higher net photosynthetic rate than female trees at elevated $CO_2$ concentrations[41]. Therefore, Salicaceae species are also an excellent model to explore the evolution of sexual dimorphism. Sexual dimorphism has been reported for morphology and reproductive investment in catkins in *P. euphratica*[43,44]. However, the genomic, transcriptomic, and epigenetic basis for this sexual dimorphism remains unclear.

The study of sex-chromosome evolution is essential to understand not only the molecular basis of sex determination but also the adaptive evolution of sexual dimorphism in response to various environments[38,45,46]. However, the dissection of these molecular mechanisms has been impeded by the lack of high-quality genomic, transcriptomic, and epigenetic data for male and female *P. euphratica* plants. In this study, we resolved the X and Y chromosomes from *P. euphratica* by performing independent chromosome-scale de novo assemblies, annotations, and comparisons of the female and male genomes. We identified the sex-linked region (SLR) using bulked segregant analysis (BSA) mapping approach. Further, we characterized the structural variants distinguishing the female and male SLRs, which revealed that suppression of recombination caused by a chromosomal inversion may be responsible for the emergence of the sex chromosome in *P. euphratica*. In *P. euphratica*, we noted that the master regulator in SLR is the partial duplicate of *ARR17*, whose repeated fragments generate 24-nt small interfering RNAs leading to male-biased methylation of *ARR17*. Furthermore, these *ARR17* repeat fragments were flanked by palindromic repeats that maintain the sequence integrity of the SLR. The SLR also accumulated repetitive sequences that likely contributed to the rapid expansion of male-specific Y regions (MSYs). Last, we identified two male-specific fusion genes in the SLR that may be responsible for sexual dimorphism in immune responses. Outside the SLR, we also identified other differentially expressed and differentially methylated genes between female and male *P. euphratica* plants that may contribute to the molecular basis of sexual dimorphism. Our results provide insights into the complex mechanism of sex determination, the dynamic evolution of sex chromosomes in *Populus*, and the adaptive evolution of sexual dimorphism in diecious plants.

## Results

**De novo assembly and annotation of *P. euphratica* male and female genomes**. We analyzed k-mer frequencies from 20 independent *P. euphratica* individuals, which indicated that the *P. euphratica* genome varies in heterozygosity from 0.70 to 1.99% (Supplementary Data 1–2). Male *P. euphratica* individuals are heterogametic, while female trees are homozygous for the sex

**Table 1 Characteristics of the female and male genomes of _P. euphratica_.**

| Category | Summary statistics | |
|---|---|---|
| Sample (sex) | TF1–136 (Female, XX) | TM7 (Male, XY) |
| Estimated genome size (k-mer analysis) (Mb) | 564.21 | 542.59 |
| Estimated heterozygosity (k-mer analysis) | 0.86% | 1.99% |
| Assembling strategy | Illumina + PacBio + Hi-C | Illumina + PacBio + Hi-C |
| Sequencing data (Gb) | 56.61 + 50.85 + 67.16 | 59.76 + 53.05 + 66.37 |
| Number of contigs | 432 | 3,347 |
| Number of scaffolds | 101 | 109/756* |
| Contig N50 (bp) | 2,039,164 | 892,045 |
| Scaffold N50 (bp) | 23,893,804 | 24,550,255/22,951,477* |
| Longest contig (bp) | 6,510,253 | 5,317,950 |
| Longest scaffold (bp) | 60,098,320 | 63,969,725 |
| Assembled genome size (Mb) | 511.25 | 1,032.28 |
| BUSCO | 92.5% | 95.9% |
| GC (%) | 32.29 | 32.23 |
| Repeat (%) | 64.41 | 52.38 |
| Predicted gene number | 36,792 | 70,370 (32,848/37,522) |

*All data related to scaffolds were counted based on the phased male genome TM7.1 and TM7.2 (MG).

chromosome[21]. Because of their low heterozygosity (0.86%), we employed all sequencing reads derived from TF1–136 individual (XX) (Supplementary Fig. 1a) for female genome assembly, while we reserved TM7 (XY) individual (Supplementary Fig. 1b) with relatively high heterozygosity (1.99%) for male genome assembly and haplotype phasing. To obtain high-quality male and female genome assemblies, we combined Illumina short-read and PacBio long-read sequencing data, with scaffolding informed by chromosomal conformation capture (Hi-C).

We obtained an estimated female genome (FG) size of 564.21 Mb, according to our genome survey (Table 1). We generated 50.85 Gb (90.11× coverage) of data with the PacBio platform and 56.61 Gb (100.33× coverage) of data by Illumina, which we then assembled with the Falcon software. The primary assembly FG consisted of 432 contigs with an N50 of 2039 kb, which was higher than that (900 kb) of the previously reported genome[24] (Table 1 and Supplementary Data 3). We successfully assigned 97.5% of the primary assembly into 19 chromosomes, with 2,478,833 unique di-tags from Hi-C data (Supplementary Data 4, 5). The pseudo-chromosomal female genome (FG) was 511.25 Mb in length, including 101 scaffolds with an N50 of 23.89 Mb (Fig. 1 and Table 1).

To assemble the male genome, we generated 53.05 Gb of PacBio (97.77× coverage) long reads and 59.76 Gb (110.14× coverage) of Illumina short reads. We estimated a genome size of 542.59 Mb for the male genome (Table 1 and Supplementary Data 3). The primary assembly consisted of 3347 contigs with an N50 of 892 kb and a total length of 1.03 Gb, which we divided into curated primary contigs and reassigned haplotigs after analysis by Purge haplotigs. We successfully anchored both primary contigs and reassigned haplotigs to 19 chromosomes using the Hi-C data, thus generating two assemblies: TM7.1 and TM7.2 (Supplementary Data 4, 5). The TM7.1 assembly was 505.33 Mb in length, with 109 scaffolds with an N50 of 24.55 Mb. The TM7.2 assembly was 527.14 Mb, comprising 756 scaffolds with an N50 of 22.95 Mb (Table 1 and Supplementary Data 5). By BSA analysis, we determined that chromosome 14 is the sex chromosome of _P. euphratica_ (see below). In addition, chromosome 14 in the TM7.1 assembly showed high colinearity with the X chromosome from FG (Supplementary Fig. 2), validating the quality of the assembled maternal and paternal sex chromosomes. By contrast, chromosome 14 in the TM7.2 assembly was longer than the X chromosome from either TM7.1 or TF1–136

(Supplementary Data 5) and was regarded as the Y chromosome of _P. euphratica_. Based on the assembled continuity (Supplementary Data 5), we selected the chromosome 14 of FG as the X chromosome for further analysis.

We confirmed the accuracy of the FG and male genome (MG, TM7.2) assemblies with their normalized contact matrices by Hi-C (Supplementary Fig. 3). Benchmarking Universal Single-Copy Orthologs (BUSCO)[47] analyses showed that 92.5 and 87.9% of BUSCO genes are complete in the FG and MG, respectively (Supplementary Data 6). Both the FG and MG _P. euphratica_ assemblies displayed good synteny with the _P. trichocarpa_ genome (V3.1) (Supplementary Fig. 2). These data suggested that our de novo assembled genomes for female and male _P. euphratica_ plants are of high quality (Fig. 1).

We detected a higher proportion of repeat sequences in the FG assembly compared to the previously published _P. euphratica_ genomes (64.4 vs. 44% in v1.0, 56.95% in v2.0)[24,48]. Repetitive sequences covered 329.30 Mb in the FG (Table 1), including 322.68 Mb of interspersed repeat sequences. The predominant type of transposable elements (TEs) was long terminal repeat (LTR) retrotransposons, with a total length of 297.10 Mb, mainly consisting of 249.42 Mb from the LTR/Gypsy and 24.51 Mb from the LTR/Copia. We obtained a similar proportion of repeat elements in the MG assembly (Supplementary Data 7). The distribution of repeat sequences in the chromosomes showed a negative correlation with gene density and a positive correlation with DNA methylation levels (Fig. 1).

We annotated the two genome assemblies with a combination of ab initio gene predictions and homology searches, which we then integrated with transcriptome deep sequencing (RNA-seq) data from leaves, petioles, stems, and catkins for each sex, resulting in the prediction of 36,792 and 37,522 protein-coding genes in the FG and MG assemblies, respectively (Table 1). We obtained functional annotations for 94.3 and 94.9% of predicted genes for the FG and MG assemblies, respectively, from at least one of the following databases: non-redundant (NR), Swiss-Prot, Kyoto Encyclopedia of Genes and Genomes (KEGG), InterPro, Pfam, and Gene Ontology (GO) (Supplementary Data 8). In addition, we identified 501 microRNAs (miRNAs), 642 transfer RNAs (tRNAs), 238 ribosomal RNAs (rRNAs), and 754 small nuclear RNAs (snRNAs) in the FG assembly, and 846 miRNAs, 757 tRNAs, 1101 rRNAs, and 551 snRNAs in the MG assembly (Supplementary Data 9). These results reflected the good

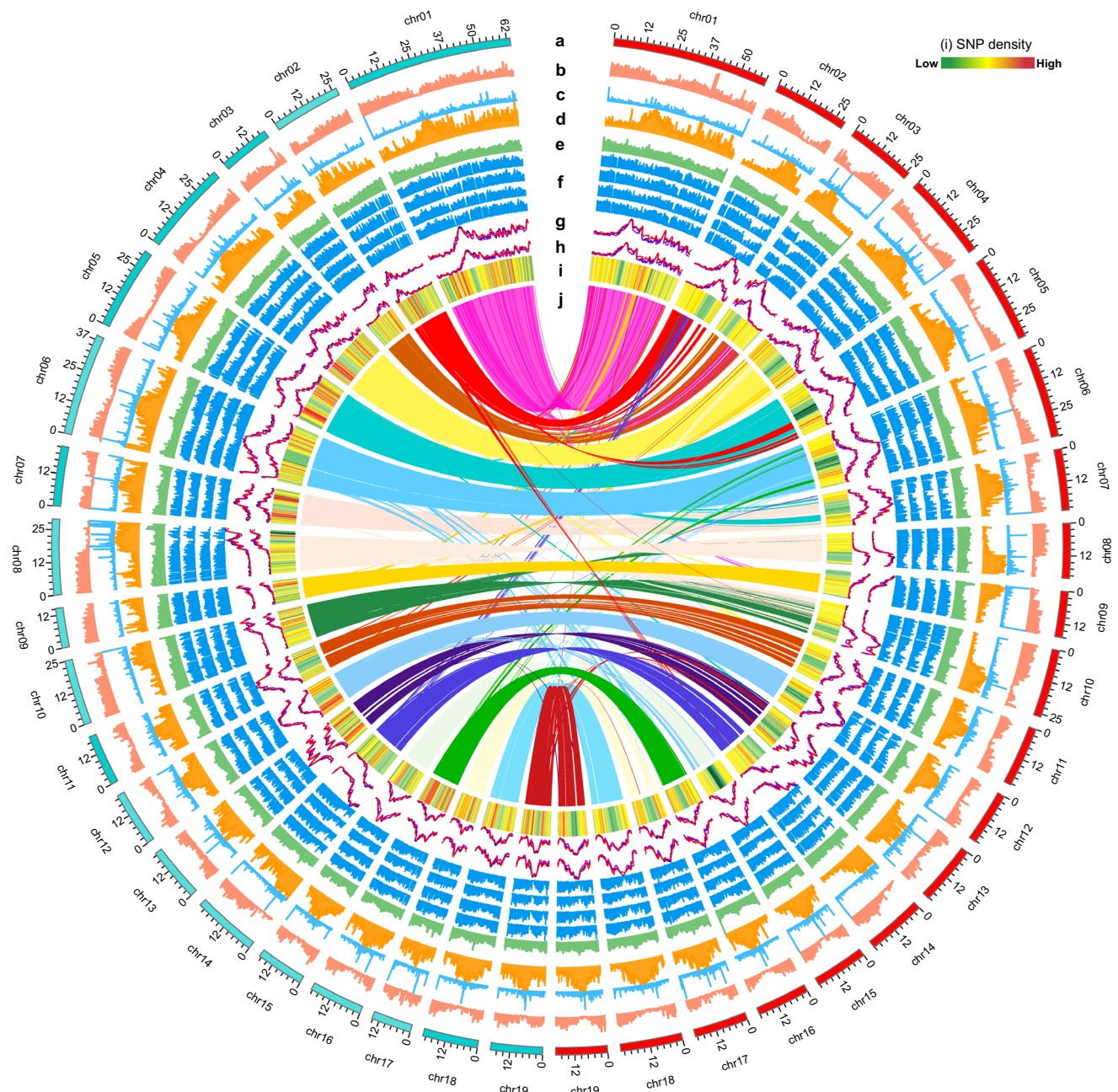

**Fig. 1 Chromosomal-scale male and female *P. euphratica* genomes with the integration of diversity, expression, and methylation data. a** Circular representation of 19 pseudomolecules from the male (cyan, left) and female (red, right) genomes. **b–e** Density of genes (**b**), tandem repeats (**c**), transposons (**d**), and GC contents (**e**). **f** Gene expression levels in catkins and leaves from female and male plants (from outer to inner tracks: female catkin, female leaf, male catkin, and male leaf). **g**, **h** DNA methylation levels in the CG (red), CHG (blue), and CHH (purple) contexts in young stems from female (**g**) and male (**h**) plants. **i** SNP density generated by BSA (color gradient key). **j** Syntenic relationships between the male and female genomes.

completeness of the annotations of the two assemblies and suggested that the MG encodes more genes than the FG in *P. euphratica*.

**Identification of the SLR on chromosome 14 of *P. euphratica*.** To identify the SLR in *P. euphratica*, we used bulked DNA pools from 100 female trees (BSA-F) and 96 male trees (BSA-M) collected in the wild to perform BSA (Supplementary Data 1). We thus generated a total of 67.17 Gb (~131× coverage) and 72.74 Gb (~138× coverage) of clean data from female and male pools, respectively (Supplementary Data 10). We mapped the BSA-F and BSA-M reads to both the female and male reference

genome assemblies to detect sex-associated SNPs (Supplementary Data 11).

We detected 3,955,478 and 6,063,291 effective SNPs against the FG and MG assemblies, respectively (Supplementary Data 12). We identified the candidate SLR based on the association of SNPs with the sex phenotype. Autosomal regions exhibit a balanced number of male-specific and female-specific SNPs, whereas a putative SLR displays an overrepresentation of male- or female-specific SNPs (above the 95% confidence threshold). When we used MG as the reference genome, we located candidate SLRs in the peritelomeric regions of chromosome 5, between 31,845 kb and 31,849 kb (4 kb in length), and of chromosome 14, between

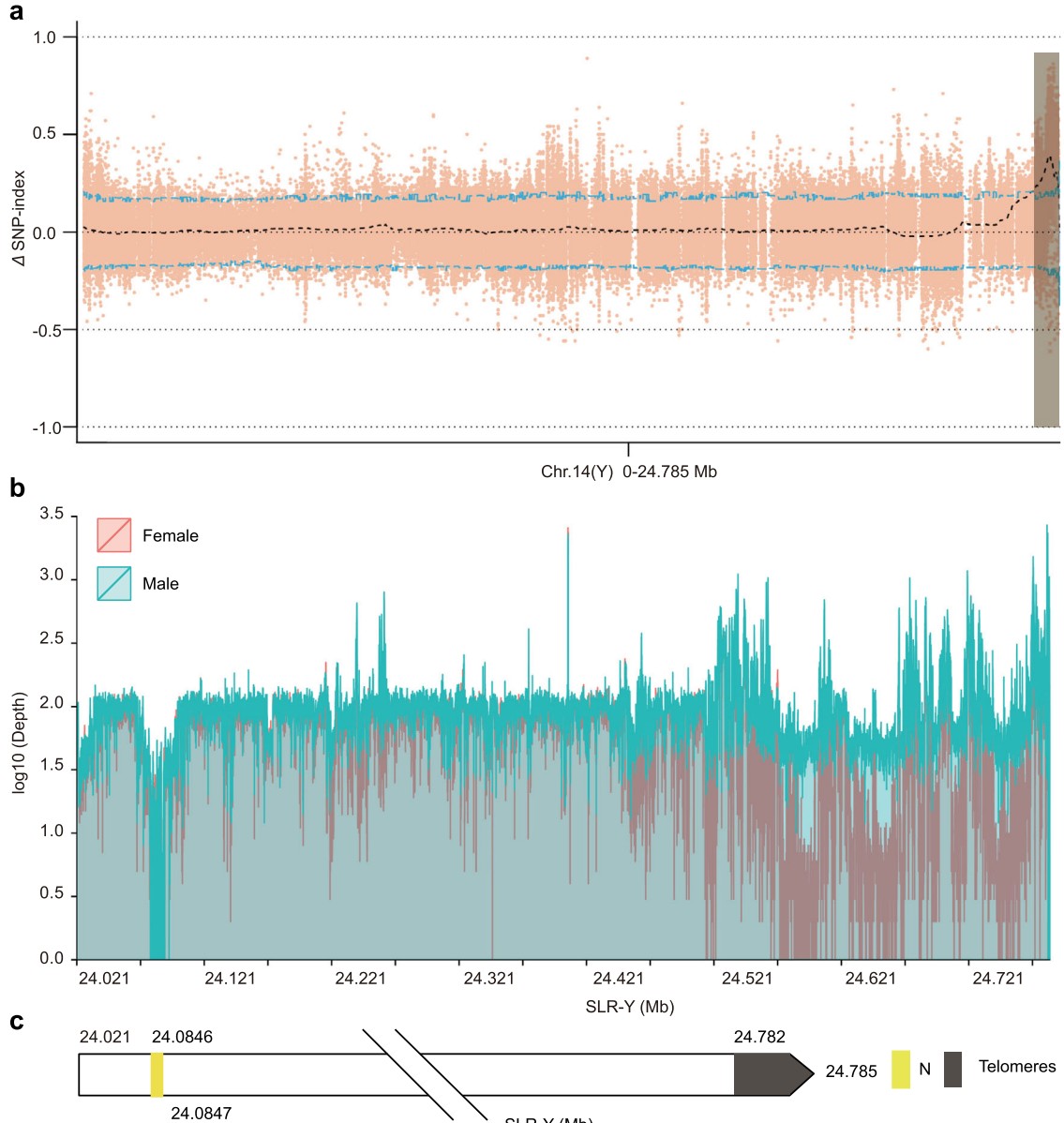

**Fig. 2 Identification and characterization of the SLR in *P. euphratica*. a** Candidate SLR (indicated by gray bar) located in the peritelomeric region of chromosome 14 (Y), as determined by ΔSNP_index analysis. **b** Differential depth of coverage across SLR-Y in the male genome using male and female BSA data, respectively. **c** Genomic diagram of SLR-Y with one distal gap represented by "N" (yellow) and telomere.

24,021 kb and 24,782 kb (761 kb in length) (Supplementary Data 13 and Supplementary Fig. 4). When using FG as the reference, we obtained two candidate SLRs: one on chromosome 3 between 25,989 kb and 25,990 kb (1 kb in length), 22,654–22,656 kb (2 kb) and one peritelomeric region on chromosome 14 between 22,677 kb and 22,684 kb (7 kb in length) (Supplementary Data 13 and Supplementary Fig. 5). As the SLR should only exist in MG (XY) but not in FG (XX), we hypothesized that the true SLR should be longer than 7 kb, leaving only the peritelomeric region of 761 kb on chromosome 14 in MG as the SLR for *P. euphratica* (Fig. 2a and Supplementary Data 13). The read coverage over this region in BSA-F and BSA-M supported its status as candidate SLR in MG (Fig. 2b). As the male SLR region (SLR-Y) was close to the end of the chromosome, we were pleased to note the intact telomere assembled near SLR-Y on chromosome 14 without any gaps. In addition, the 761-kb SLR-Y comprised two contigs, while the

corresponding region on X (SLR-X) was included within a single contig (Fig. 2c). Together with the synteny analysis between the SLR-Y and the reported sex-determining region[21] of *P. euphratica* (Supplementary Fig. 6), these results suggested that the peritelomeric SLR-Y identified here is relatively complete.

**Structural variation and evolution of the SLR**. By syntenic analysis, we identified four large inverted fragments between the X and Y chromosomes from the FG and MG assemblies (Fig. 3a), but only one inversion (from 22,374 kb to 22,616 kb in X and from 24,215 kb to 24,453 kb in Y) located within the SLR. We confirmed the existence of this inversion in one complete contig bearing the fragment in sex chromosomes (Supplementary Data 14); in addition, PacBio long reads spanned two breakpoints of the inverted repeat (Supplementary Fig. 7).

Besides this inverted fragment, a sequence alignment between SLR-Y and its counterpart region on the X chromosome from the

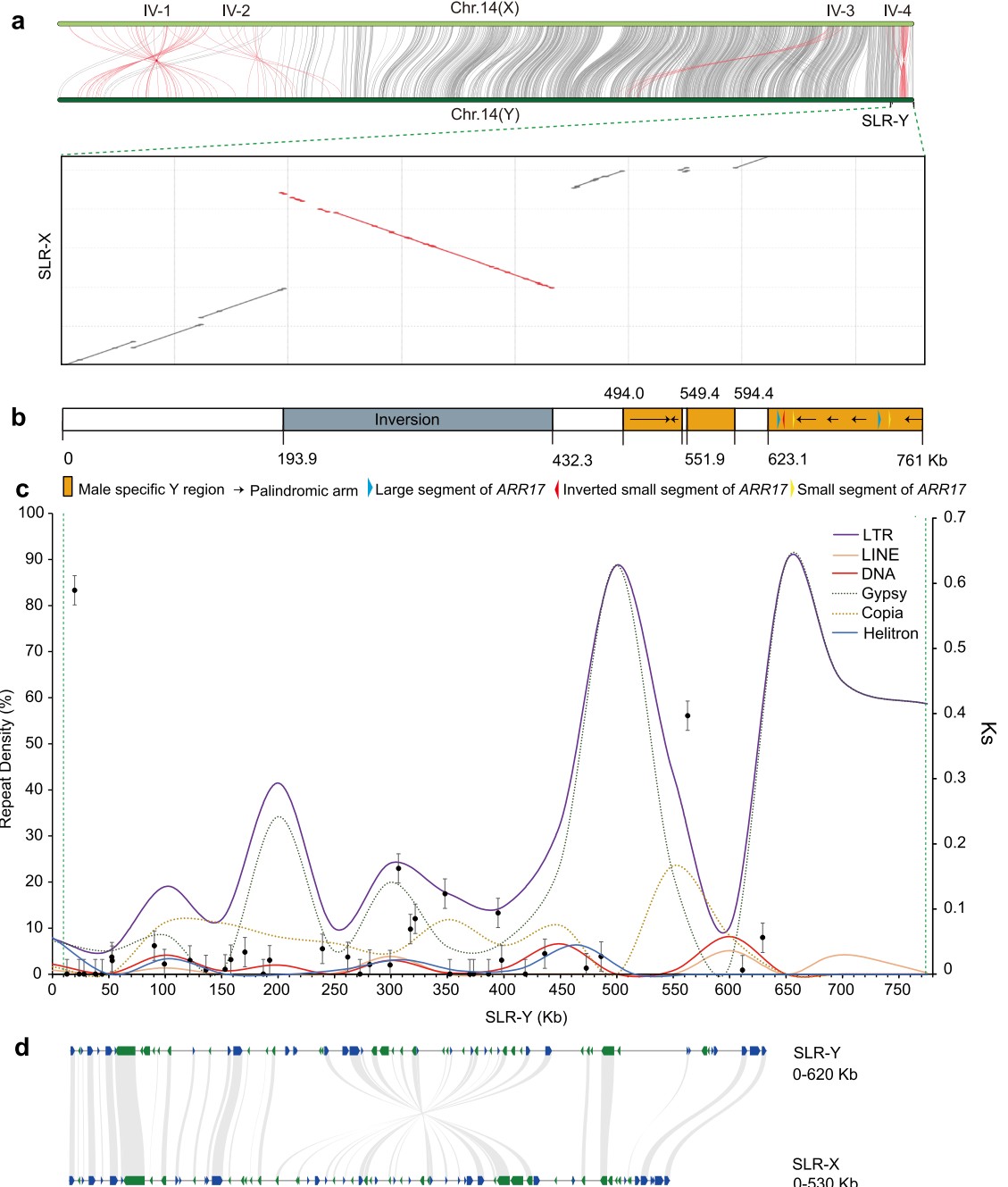

**Fig. 3 Structural and sequence characteristics of SLR on chromosome 14 in *P. euphratica*. a** Synteny plot between the X and Y chromosomes, including SLR-Y and the counterpart region SLR-X. Four inversed fragments (IV-1, 2, 3, and 4) were marked in red. **b** Structural variation at SLR-Y. **c** Repeat types and repeat density in SLR-Y, and Ks of homolog gene pairs between SLR-Y and SLR-X. **d** Colinearity between genes from SLR-Y and SLR-X.

FG assembly (SLR-X) also indicated that SLR-Y contains additional characteristic structures important for sex-chromosome evolution in *P. euphratica* (Fig. 3b). In particular, we identified three MSYs with lengths of 55.4 kb (MSY-I), 43.3 kb (MSY-II), and 137.6 kb (MSY-III) (Supplementary Data 14). Some fragments with high sequence identity to the master regulator *ARR17* were located in MSY-III (Fig. 3b). We also detected six palindromic arms (arm1–6) located within MSY (Fig. 3b and Supplementary Fig. 8), indicating that these *ARR17* fragments are surrounded by inverted repeats, which may play a role in maintaining sequence integrity of the SDR[17] (Fig. 3b). We also observed high density of repetitive elements specifically accumulated within MSYs (Fig. 3c). MSY-I consisted of 91.2%

repetitive sequences, all in the form of LTR/Gypsy. MSY-II contained 48.5% repetitive sequences, of which 74.9% was LTR/Copia and 22.3% LTR/Gypsy. MSY-III was characterized by 74.8% repetitive sequences, 92.9% of which were LTR/Gypsy (Supplementary Data 15). The proportion of repeats was thus clearly higher than 43.2% for the entire SLR-Y, compared to 29.1% for SLR-X. The emergence of MSYs with high repeats indicated that the accumulation of repetitive sequences probably contributed to the rapid expansion of SLR-Y in *P. euphratica*.

The comparison of homologous gene pairs between SLR-Y and the counterpart region SLR-X (Fig. 3d) allowed us to investigate the relationship between SLR divergence and the occurrence of the inversion within SLR. We thus calculated the synonymous

substitution rate (Ks) of 39 gene pairs between SLR-Y and SLR-X and 17 gene pairs located in the inversion (Fig. 3d). After removing outliers, the average Ks for genes within the SLR was 0.018, which was lower than the Ks value of 0.040 obtained for the inversion (Fig. 3c). Using the divergence time between *Populus* and *Salix* 60 million years ago (Mya)[25] and the average Ks of 0.0146 between *P. trichocarpa* and *S. purpurea*[17] as references, we estimated the divergence of SLR-X and SLR-Y within *P. euphratica* to have occurred about 7.46 Mya, later than the inversion, which dated from 16.44 Mya. This result indicated that the inversion event took place before the recombination suppression of SLR and thus may have contributed to the evolution of sex chromosomes by extending the non-recombining region.

**Sex-biased DNA methylation at *ARR17*.** The feminizing factor *ARR17* in *Populus* shows a female-specific expression pattern in the early stages of flower bud formation[6,21,22]. We performed a sequence search using each exon of *ARR17* (*PeuTM19G01068*) as a query in *P. euphratica*, which detected two large homologous segments (L1 and L2) and three small homologous segments (S1–S3) within the MSY-III region (Fig. 4a and Supplementary Data 16). Small RNA data generated from early flower buds[21] showed that small RNA reads, especially 24-nt RNA reads (>75%), specifically map to the small S1 and S2 segments (Supplementary Fig. 9), suggesting that the *ARR17* fragments in MSY may regulate the transcription or transcript levels of their target gene *ARR17* via the formation of 24-nt non-coding RNAs (ncRNAs). Our results thus supported the view that silencing of *ARR17* in male trees is mediated by RNA-directed DNA methylation (RdDM)[6], in which 24-nt small interfering RNAs (siRNAs) induce de novo DNA methylation in their target loci. In contrast to its paralogous gene *ARR16* (*PeuTM19G00529*) encoding a protein with a similar domain (Fig. 4b), we only observed sex-specific hyper-methylation of the *ARR17* promoter in plants grown in two independent habitats (Fig. 4c and Supplementary Data 17). Furthermore, the genomic regions with homology to exon 1 within four segments (S1–S3, L2) exhibited evidence of relaxed selection with a Ka/Ks ratio >1 (Fig. 4d and Supplementary Data 18), possibly helping these male-specific *ARR17* fragments function as ncRNAs. A hairpin RNA structure formed only between the inverted S1 and S2, but not between S3 and L2, indicating an important role for inverted repeats in small RNA formation[6,49]. These results suggested that the formation of 24-nt siRNAs may have resulted from the convergence of relatively relaxed selection and the hairpin structure between S1 and S2 in male-specific *ARR17* fragments within the MSY-III region, leading to specific repression of *ARR17* transcription via the RdDM pathway in male *P. euphratica* trees.

In addition to *ARR17*, we identified several other genes with sexually dimorphic methylated sites in their putative promoter regions, which might contribute to the sexual dimorphism of other traits. We detected 840 and 327 genes for male-specific and female-specific hypermethylated genes in young stems of *P. euphratica*, respectively (Supplementary Fig. 10). A GO term enrichment analysis for biological processes of these male hypermethylated genes revealed that they are mainly enriched in tryptophan and tyrosine biosynthesis and in metabolic processes involving organic substances including cellular biogenic amine, lipids, L-phenylalanine, starch, sucrose, and purine nucleobases (Fig. 4e). Female hypermethylated genes were mainly enriched in physiological responses such as "proteolysis", "oxidation-reduction", "phosphorylation", "transmembrane transport" and "regulation of transcription" (Fig. 4f and Supplementary Data 19). This result suggested that these

epigenetic modifications may affect other biological processes and thus contribute to other aspects of sexual dimorphism in *P. euphratica*.

**Sex-biased expression of genes and two fusion genes created by fragment inversion in the SLR.** Among the 39 orthologous genes in SLR-Y and SLR-X, some genes related to reproductive tissue showed sex-biased expression (Fig. 5a and Supplementary Data 20). For example, the gene homologous to Arabidopsis (*Arabidopsis thaliana*) *WINDHOSE2* (*WIH2*), *PeuTF14G01779* (in FG), and *PeuTM14G01689* (in MG), was expressed 11 times more highly in female catkins than in male catkins. In Arabidopsis, WIH2 localizes to the plasma membrane and participates in megasporogenesis during the transition from somatic to reproductive cell fate[50]. Another example was *PeuTM14G01707*, which exhibited male-specific expression in catkins, in line with the function of its Arabidopsis homologs *PRK4* (*POLLEN RECEPTOR-LIKE KINASE 4*) involved in protein phosphorylation during pollen germination and tube growth[51,52]. In fact, several gene pairs had the best hits to genes related to pollen development in Arabidopsis by BLAST and showed biased expression in male catkins, including the pairs *PeuTM04G01684* and *PeuTF14G01785*, *PeuTM14G01691* and *PeuTF14G01777*, *PeuTM14G01694* and *PeuTF14G01772*, and *PeuTM14G01696* and *PeuTF14G01768*. The best hit for *TRANSLOCASE OF THE OUTER MITOCHONDRIAL MEMBRANE 40* (*TOM40*) in *P. euphratica* (*PeuTM14G01682* and *PeuTF14G0787*) exhibited a 2.6-fold higher expression in male leaves than in female leaves. TOM40 is a core subunit of the TOM complex and is required for mitochondrial biogenesis and embryo development[53].

Several genes related to stress response and growth also displayed a sex-biased expression pattern. For example, *PeuTM14G01696* expression levels were five times higher in male catkins than in female catkins (its matching gene being *PeuTF14G01768*); their best BLAST hit was Arabidopsis *ACR1* (*ACT DOMAIN REPEAT 1*), whose expression is downregulated by salt stress and upregulated by cold[54]. *PeuTM14G01684* exhibited male-biased expression (male/female ratio >4) in catkins and was a putative ortholog for Arabidopsis *TOP1* (*THIMET METALLOENDOPEPTIDASE 1*). TOP1 enzymatic activity is blocked by binding of the defense phytohormone salicylic acid and is a necessary component of plant effector-triggered immunity in response to pathogen infection[55,56]. *PeuTM14G01669* showed a female-limited expression in catkins and a male-biased expression in leaves; its best BLAST hit was with Arabidopsis *GRF1* (At2g22840, *GROWTH-REGULATING FACTOR 1*) encoding a master regulator that modulates the balance between stress and defense response and plant growth[57]. *PeuTM14G01719* was expressed five times more highly in female leaves than in male leaves; it was a putative ortholog of Arabidopsis *HCF173* (*HIGH CHLOROPHYLL FLUORESCENCE PHENOTYPE 173*) whose encoded protein is involved in translation initiation of the *psbA* mRNA and photosystem II assembly[58]. These genes with sex-biased expression in the SLR may contribute to secondary sexual dimorphisms in development and stress responses to salt, cold, and disease, as well as their balance.

Besides biased expression observed for the syntenic genes, we noticed sex-specific genes within the SLR that may contribute to sexual dimorphism in disease resistance, possibly playing a key role in the development of reproductive organs. For example, the SLR-Y-specific genes *PeuTM14G01677* and *PeuTM14G01678* were located at the left breakpoint of the inverted region in SLR-Y (Fig. 5b) and showed a best hit with Arabidopsis *RPPL1* (putative disease resistance *RPP13-LIKE PROTEIN 1*, At3g14470)

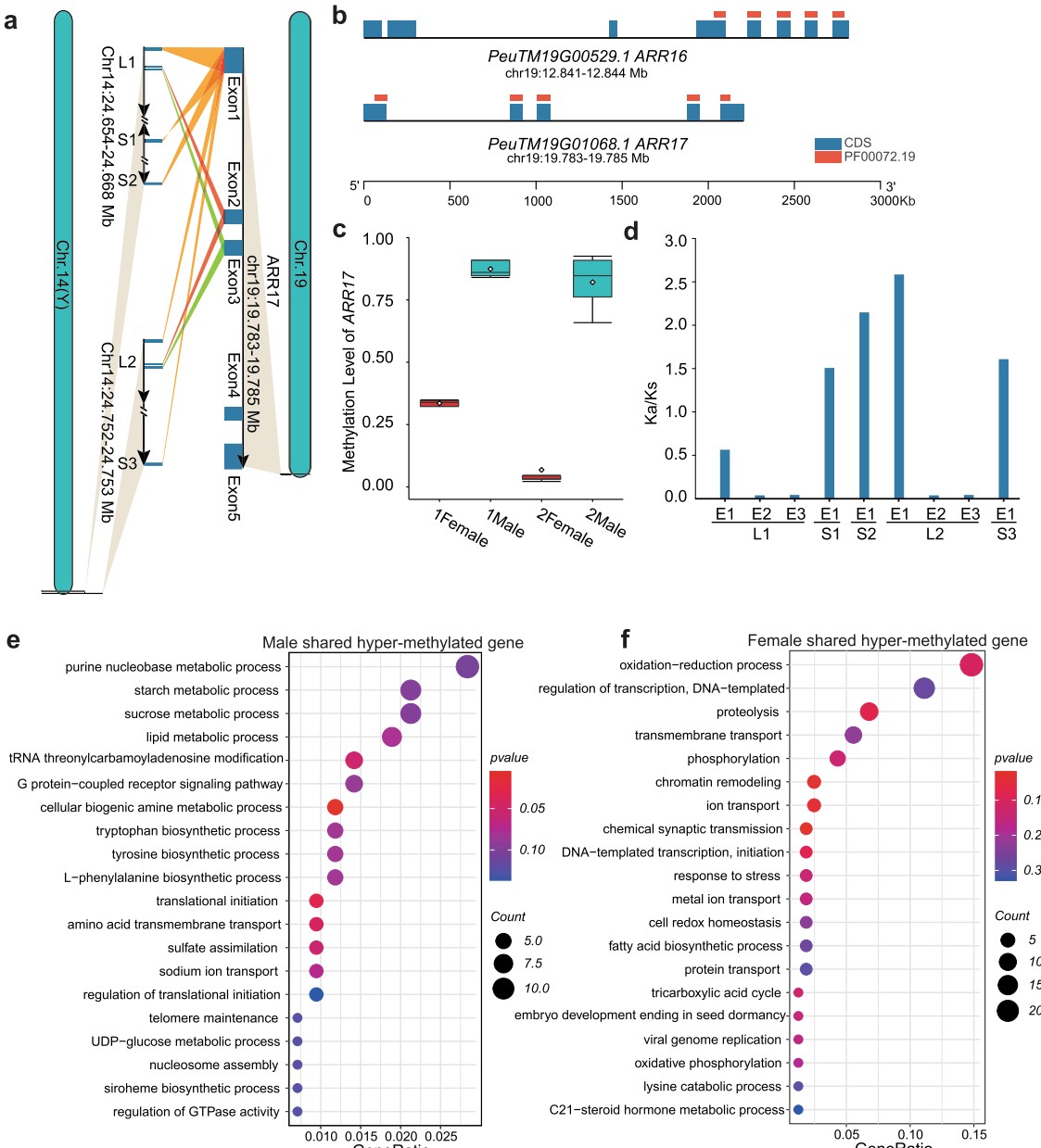

**Fig. 4 Differential methylation levels of ARR17 in males and females contributing to the sex determination in *P. euphratica*. a** *ARR17* on chromosome 19 is partially duplicated in MSY-III of chromosome 14 (Y). **b** Schematic representation displaying the coding sequences of *ARR17* and its paralogous gene *ARR16* on chromosome 19 of *P. euphratica*. **c** Methylation levels at the *ARR17* locus in young stems from female and male *P. euphratica* plants in different environments. 1 and 2 represent samples from two different environments. In the box plot, center lines indicate the median, box limits indicate the first and third quartiles. **d** Analysis of the selective pressure on *ARR17* exons, as represented by Ka/Ks ratios. **e, f** Separate GO term enrichment analysis for male-specific and female-specific hypermethylated genes (biological processes).

encoding an NB-ARC (nucleotide-binding adapter shared by Apaf1, certain R proteins and CED4) domain (Supplementary Data 21). Their best hits in the *P. trichocarpa* genome were *Potri.014G003600* and *Potri.004G170232*, respectively, both of which encode members of the resistance gene analog (RGA) family with a nucleotide-binding site sharing homology to eukaryotic cell death effectors. Further analysis showed that *PeuTM14G01677* is a fusion gene located at the breakpoint of the inversion in SLR-Y, consisting of a partial colinear fragment with SLR-X and a partially inverted fragment, while *PeuTM14G01678* was composed of a partial inversed segment and an unknown region (Fig. 5b). Phylogenetic and domain analysis of the two encoded fusion proteins and their homologs in the MG and FG

showed that the two male-specific genes group into two clades, containing an NB-ARC domain and/or a leucine-rich repeat (LRR) domain. Based on our RNA-seq data in leaves and catkins, each of the eight *RGA* genes in male and female trees reached their highest expression in male leaves (Fig. 5c), suggesting an important role for these genes in male leaves. The two male-specific genes (*PeuTM14G01677* and *PeuTM14G01678*) generated by the inversion were the most highly expressed *RGA* genes in leaves. We only identified two similar *RGA* genes in female trees: *PeuTF03G00210* with no differential expression and *PeuTF03G00209* encoding a protein lacking an NB-ARC or LRR domain, suggestive of the impairment or loss of RGA function in female trees. Field observations also indicated that

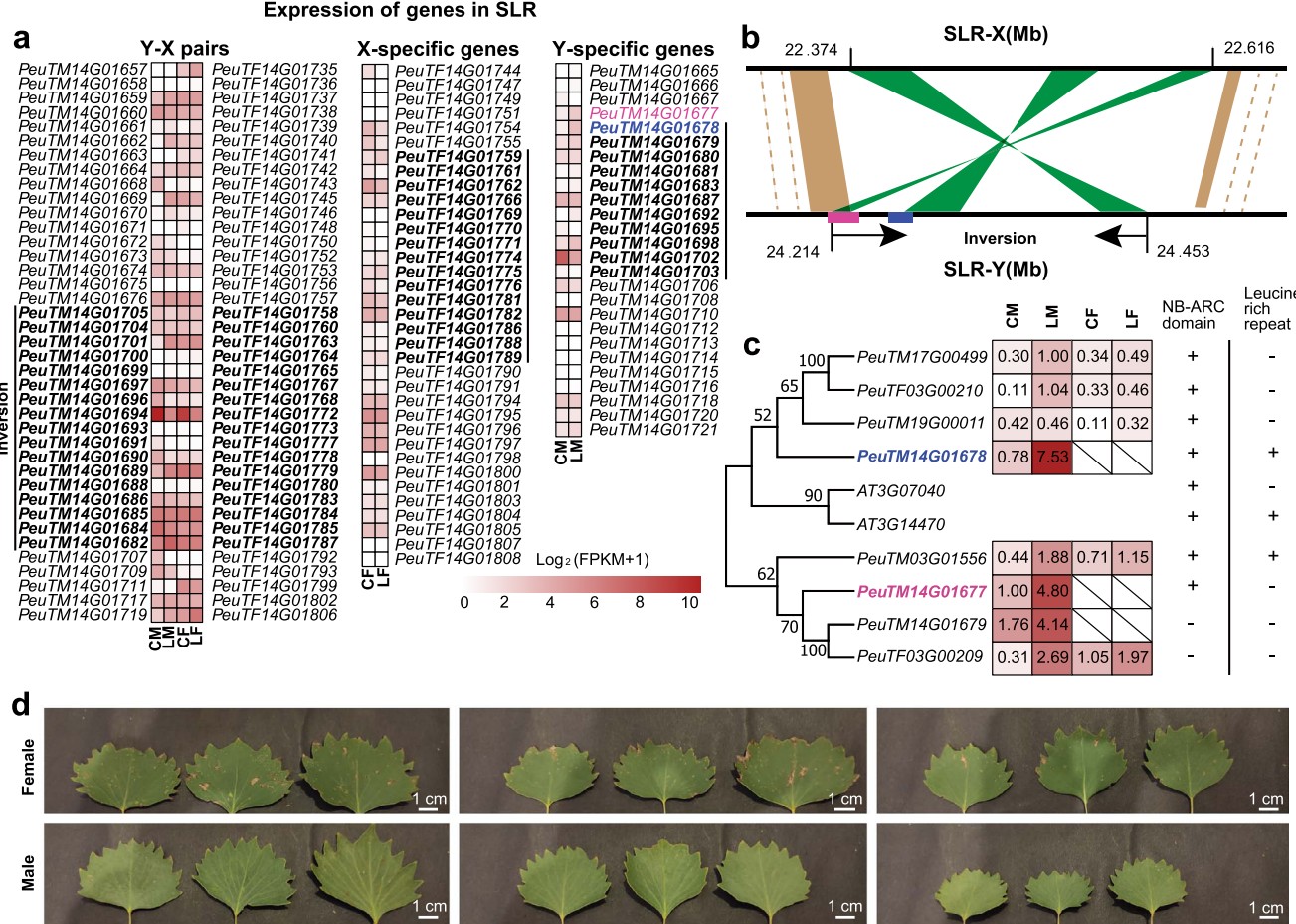

**Fig. 5 Gene contents in the SLR and sexual dimorphism in resistance to disease. a** Heatmap representation of gene expression for SLR-X genes and their counterparts from SLR-Y, SLR-X-specific, and SLR-Y-specific genes. Genes located within the inverted region are highlighted in bold. **b** Gene model of the fusion gene *PeuTM14G01677* (red) and the neighboring gene *PeuTM14G01678* (blue), *RGA*, in the inverted boundary between SLR-X and SLR-Y.
**c** Phylogeny, expression levels and domains of encoded RGAs in the female and male genomes of *P. euphratica*. The numbers in the box represent FPKMs. For (**a**, **c**), CM catkin male, CF catkin female, LM leaf male, and LF leaf female. **d** Differential resistance of male and female leaves in response to fungal disease in three wild male and three wild female *P. euphratica* trees. Scale bar, 1 cm.

male trees exhibit a stronger resistance against disease than female trees (Fig. 5d). In addition, the two SLR-X-specific genes *PeuTF14G01754* and *PeuTF14G01755* (Supplementary Data 22), located immediately adjacent to the inversion in SLR-X, showed higher expression levels in female catkins than in female leaves (Fig. 5a). Their best hit was Arabidopsis *ANT* (*At4g37750*, *AINTEGUMENTA*), which is one of the key genes involved in the positive regulation of auxin biosynthesis, with a role in gynoecium development[59,60]; *ant* mutants exhibit reduced fertility and abnormal ovules[61].

**Sexually dimorphic gene expression.** Sexual dimorphism in gene expression was not specific to reproductive tissues, as we also detected sex bias in vegetative tissues for *P. euphratica*. Of the 27,501 genes expressed in leaves and catkins for the male and female *P. euphratica* trees (Supplementary Data 23), we identified 2951 sex-limited genes (SLGs) (Fig. 6a), comprising 331 male-limited genes specific to leaves, 1367 male-limited genes specific to catkins and 136 male-limited genes in both tissues. For female-limited expressed genes, we obtained 286 genes that have a specific expression to leaves, 680 genes specific to catkins and 151 genes in both tissues. Furthermore, we detected 6,379 sex-biased genes (SBGs), consisting of 3135 male-biased genes and 3127

female-biased genes specific to catkins, 44 male-biased genes and 60 female-biased genes specific to leaves, and six male-biased genes and seven female-biased genes in both tissues (Fig. 6b and Supplementary Data 24). We performed GO enrichment analysis on these SLGs and SBGs in catkins and leaves.

For SLGs expressed in catkins, 1,367 male-limited genes were significantly enriched in energy-related processes, such as cation transport ($P = 3.02 \times 10^{-9}$, 24/125), starch metabolism ($P = 1.25 \times 10^{-8}$, 34/245), sucrose metabolism ($P = 1.39 \times 10^{-8}$, 34/246), response to stress ($P = 6.75 \times 10^{-5}$, 19/149), cell-wall modification ($P = 1 \times 10^{-4}$, 10/54), asexual sporulation ($P = 2 \times 10^{-3}$, 5/21) and exocytosis ($P = 6 \times 10^{-3}$, 6/36) (Fig. 6c and Supplementary Data 25). Biological processes of 680 female-limited genes were specifically enriched in responses to stress, such as response to wounding ($P = 4.97 \times 10^{-5}$, 8/58), auxin ($P = 6 \times 10^{-3}$, 7/97), and oxidative stress ($P = 3 \times 10^{-2}$, 7/132) (Fig. 6d and Supplementary Data 25). These results indicated that the different functions of these SLGs may contribute to catkin development in male and female trees.

For SLGs in leaves, we obtained the enriched GO categories regulation of translation ($P = 1.6 \times 10^{-8}$, 9/63) and protein ubiquitination ($P = 3 \times 10^{-2}$, 8/371) as being specific for the 331 male-limited genes, while the categories of lipid metabolism ($P = 2 \times 10^{-2}$, 5/215), starch metabolism ($P = 4 \times 10^{-2}$, 5/245)

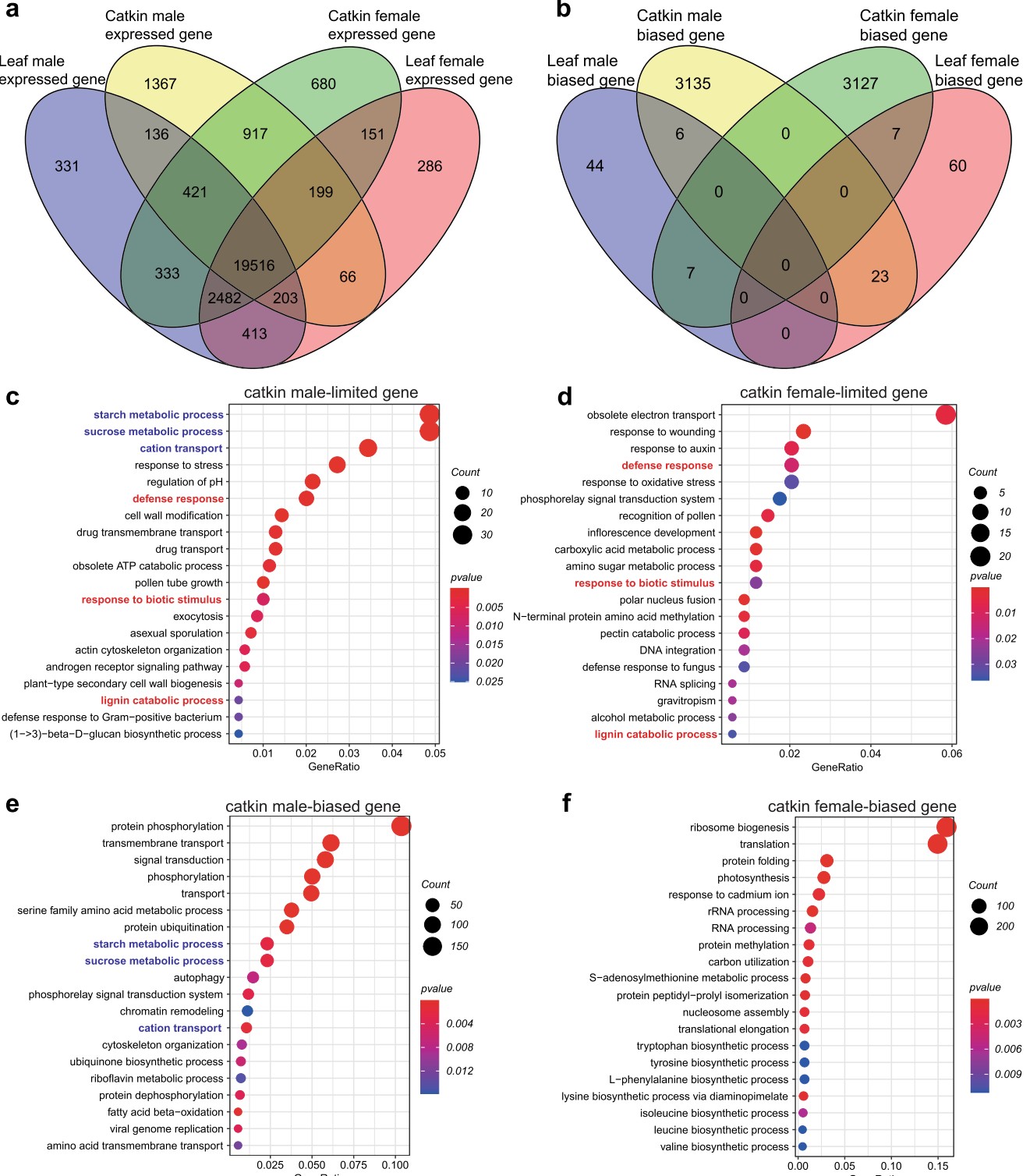

**Fig. 6 Genes differentially expressed by sex across the genome of *P. euphratica*. a** Venn diagram showing the comparison of genes expressed in catkins and leaves from *P. euphratica* male or female trees. **b** Venn diagram showing the comparison of sex-biased expressed genes in *P. euphratica* catkins and leaves. Top 20 enriched biological processes of catkin male-limited genes (**c**) and female-limited genes (**d**). Top 20 enriched biological processes of catkin male-biased genes (**e**) and female-biased genes (**f**). For **c–f**, the shared terms enriched for female- and male-limited genes are indicated in red, while the shared terms enriched for male-limited genes and -biased genes in catkin are shown in blue.

and sucrose metabolism ($P = 4 \times 10^{-2}$, 5/246) were enriched in the 286 female-limited genes (Supplementary Data 26). Although the enrichment of SLGs shared between catkins and leaves may indicate common sexual dimorphism in these two tissues in *P.*

*euphratica*, we identified very few corresponding genes (Supplementary Data 27).

Of the 21,053 genes expressed by female and male catkins, 3135 male-biased genes were enriched in rather different biological

processes compared to the 3127 female-biased genes. Male-biased genes were significantly enriched in starch and sucrose metabolism ($P = 2 \times 10^{-3}$, 41/62) and cation transport ($P = 1 \times 10^{-3}$,19/24) to name a few (Fig. 6e and Supplementary Data 28). However, female-biased genes were enriched for photosynthesis ($P = 2.32 \times 10^{-7}$, 52/64), response to cadmium ion ($P = 8 \times 10^{-4}$, 42/59), defense response to the bacterium ($P = 1.2 \times 10^{-2}$, 27/39), auxin-activated signaling ($P = 1.8 \times 10^{-2}$,12/15) and response to cold ($P = 3 \times 10^{-2}$,31/48) (Fig. 6f and Supplementary Data 28). Of the 22,614 genes expressed by female and male leaves, 44 male-biased genes were significantly enriched in protein phosphorylation ($P = 1.4 \times 10^{-2}$, 6/266), while 60 female-biased genes showed no significant enrichment (Supplementary Data 29).

The expressed genes showing sex bias in both catkins and leaves may be essential for sexual dimorphism (Supplementary Data 30). For example, *PeuTF05G02133* displayed biased expression in male catkins and leaves; its best hit was Arabidopsis *RD19A* (*At4g39090*, *RESPONSE TO DEHYDRATION 19A*). RD19A is involved in tapetal degradation and pollen development mediated by the vacuolar processing enzyme βVPE[62] and is required for resistance against the pathogenic bacterium *Ralstonia solanacearum*[63].

Of these genes with sexually dimorphic expression, only a few may be regulated by sex-specific methylation, as is the case with the feminizing factor *ARR17*. We evaluated methylation status at the chromatin of these genes and detected sex-specific methylation sites at the promoter regions of 2.64% (78/2951) of SLGs and 2.95% (188/6379) of SBGs (Supplementary Fig. 11). Several sex-specific methylated genes may shape sexual dimorphism. For example, the homolog for *2-Alkenal reductase* (*AER*), *PeuTF17G00028*, exhibited hypermethylation in male trees and a female-limited expression pattern. *AER* encodes a 2-alkenal reductase (EC 1.3.1.74), which plays a key role in the detoxification of reactive carbonyls and is thus involved in response to oxidative stress[64,65]. The homolog to Arabidopsis *MYB68*, *PeuTF06G02282*, displayed hypermethylation in females and a male-limited expression pattern. Overexpression of Arabidopsis *MYB68*, encoding an MYB domain transcription factor with N-terminal R2R3 DNA-binding domains, raises tolerance to heat and drought during reproductive stages[66]. These results indicated that sexual dimorphism may originate from the regulation of these genes by sex-specific methylation.

## Discussion

Sex chromosomes have evolved independently many times throughout the evolution of flowering plants, yet the underlying mechanisms of sex determination and the evolutionary dynamics of sex chromosomes in plant species have long been a puzzle. It is notoriously difficult to assemble a complete sequence for sex chromosomes, since they often contain highly repetitive sequences such as TEs and tandem gene duplications[1]. In addition, independent de novo assemblies for both the male and female genomes are seldom produced in a single study, and so direct comparative tools to dissect sex chromosomes are lacking. The implementation of high-throughput sequencing and the advent of single-molecule real-time sequencing in genome assembly have achieved substantial progress in unraveling the mechanisms of sex determination and gene contents of sex chromosomes across diverse species[6,16,22,67].

By combining Illumina, PacBio and Hi-C sequencing strategies, we de novo assembled the male and female genomes at the chromosome level for *P. euphratica*. Since the male individual TM7 has a relatively high degree of heterozygosity, we assembled most parts of the genome as separate primary contigs (haploid assembly), which produced a genome twice the size of the

predicted haploid genome (1.03 Gb), rather than a single haplotype-fused contig. We then used the draft diploid assembly and the alignment file of PacBio long-reads mapped to the assembly as inputs for the Purge haplotigs pipeline, which generated two sets of genomes (curated primary contigs and reassigned haplotigs). Compared to autosomal chromosomes, the difference between sex chromosomes was striking. We thus obtained two nearly haplotype-phased sex chromosomes for the male genome, which we confirmed against the assembled X chromosomes in the male and female genomes based on their almost perfect colinear relationship (Supplementary Fig. 2). We also obtained an independent validation with the coverage of PacBio long reads against the X and Y chromosomes (Supplementary Fig. 7).

With these two genomes as a reference, we mapped the putative SLR to chromosome 14 from a BSA of SNP frequencies in pools of male or female trees. We narrowed down the SLR to a 761-kb peritelomeric region ending with the telomere on chromosome 14 (or chromosome Y in male trees). SLR-Y is not contiguous but is interrupted by one gap. To resolve the evolution of the sex chromosome, additional work will be necessary to fill this gap or to complete the sequence of the sex chromosome from telomere to telomere, with high-fidelity sequencing technology[68,69].

The canonical single-factor models of sex-chromosome evolution in plants have proposed a roadmap: a single master regulator of sex determination switching between male and female arose, followed by an expansion of recombination suppression, chromosome differentiation, and ultimately degeneration[3,9]. According to this model, SDRs are characterized by suppressed recombination and a sex-determining factor present in only one sex.

The first step of sex-chromosome evolution in SLR-Y in *P. euphratica* was the partial duplicates of repeated fragments from *ARR17*. In this study, we attempted to infer the mechanism by which these repeated fragments arose. Emerging evidence suggests that super-families of transposons can duplicate genes or gene fragments, thus driving innovations at the molecular and phenotypic levels[70]. We, therefore, created an inventory of all TEs within SLR-Y. We observed a high density of Gypsy TEs in the region flanking the repeated fragments of *ARR17*, indicating that *ARR17* fragments may have been captured and translocated to chromosome 14 by Gypsy TEs in *P. euphratica*.

The presence of palindromes in the sex chromosomes of mammals and plants has been considered a common characteristic that helps maintain sequence identity in the absence of homologous recombination to correct errors[17,71]. We, therefore, used LASTZ to detect whether palindromic repeats by aligning the SLR-Y region to itself. We detected palindromic repeats in the regions flanking the repeated fragments of *ARR17*. Recent studies in other plant species also determined that the regions flanking the sex master regulator are male-specific or hyper-repetitive, including palindrome-like structures, suggesting that this characteristic may be common across sex-chromosome evolution in plants[72].

In many animals and plants, recombination suppression between sex chromosomes is processed in a stepwise manner, presumably through chromosomal inversions, resulting in a stratified pattern of sequence divergence between the two sex chromosomes, known as strata[73,74]. In this study, we looked for strata in the SLR by scanning different degrees of divergence based on synonymous substitutions (Ks) calculated with syntenic gene pairs identified between SLR-X and SLR-Y. Our results showed that at least two strata can be recognized in the SLR-Y: one old stratum with an average of Ks >0.1 and one young stratum with an average of Ks <0.05. The old stratum overlapped

with the inversion fragment, indicating that this inversion occurred earlier during the evolution of the sex chromosome and perhaps played an important role in suppressing recombination after the emergence of the duplicated *ARR17* fragments. We also wished to investigate the driving force behind the fragment inversion. Accordingly, we determined the density of TEs and detected the specific accumulation of Gypsy in SLR-Y, showing a higher density than any other TE type. Furthermore, we distinguished two peaks in Gypsy density for the two breakpoint regions on either side of the large fragment inversion. Therefore, we hypothesize that the large fragment inversion between SLR-Y and SLR-X was driven by Gypsy in *P. euphratica*, as Galileo and other transposons did in the fruit fly (*Drosophila melanogaster*)[75,76].

Dioecy evolved independently in different species of the genus *Populus*[22], although several species (*P. euphratica*[21], *P. deltoides*[22], *P. tremula*[6], *P. tremuloides*[6], *P. trichocarpa*[6,18], and *P. balsamifera*[6,77]) seems to share a common mechanism of sex determination: a partial gene duplication was identified in the SDR that produces siRNAs to silence the expression of a target gene[6,7,22].

We established here that the relaxed selection of exons in the duplicated segments S1–S3 may generate small ncRNAs. Among the three small segments in MSY-III, only the two inverted segments were covered by small RNA reads, particularly 24-nt siR-NAs, confirming that the partial duplications of *ARR17* in the SLR can produce 24-nt siRNAs, possibly in an RNA polymerase II-dependent manner, based on the structure of the hairpin[49,78]. siRNAs derived from an inverted repeat have also been reported in *P. tremula*[6], *P. deltoides*[22], and maize (*Zea mays*)[79]. Nonetheless, the function of the small segment S3 that also undergoes relaxed pressure remains to be explored.

24-nt siRNAs play a central role in triggering RNA-directed DNA methylation in plants[49,80]. DNA methylation is important for sex differentiation in plants. In *P. tremula*, the feminizing factor *ARR17* is an autosomal gene specifically expressed in female flower buds and targeted for methylation by a siRNA derived from the Y-specific inverted *ARR17* repeat fragments[6,81]. In *P. deltoides*, the expression of female-specifically expressed *RESPONSE REGULATOR* (*FERR*) (*ARR17* ortholog) is suppressed in male flowers, and the *FERR* promoter exhibits male-specific methylation directed by a siRNA derived from the *PdeFERR-R* locus containing multiple *FERR* segments[22]. In *P. balsamifera*, the promoter and first intron of *PbRR9* (*ARR17* ortholog) also have male-specific methylation sites[6,77]. In *Silene latifolia*, about 21% of genetically male plants can develop into androhermaphrodites when treated with a hypomethylating drug[82]. In this study, we observed male-specific hypermethylation in the CHG and CHH contexts at the *ARR17* promoter in *P. euphratica* plants exposed to two different environments. Together with its male-specific repression of expression in flower buds[21], these results indicated that DNA methylation plays an important role in sex differentiation. However, the detailed regulatory mechanism of the siRNA in the context of sex determination will need to be further explored at the epigenetic level.

In this study, we identified the two fusion *RGA* genes *PeuTM14G01677* and *PeuTM14G01678*. Gene fusions are one of the main paths to the creation of a novel gene, playing critical roles in gene evolution[83,84]. For example, a gene fusion between *E3* and *SoyZH13-19G210600* in soybean (*Glycine max*), caused by a 13.3-kb deletion, may be associated with flowering time[85]. Similarly, the high-quality assembly of sex chromosomes allows us not only to detect structural variation between the X and Y chromosomes but also to identify gene fusion events created by fragment inversions.

Sexual dimorphism in diecious plants is thought to be less common and less conspicuous than in animals[32]. For sexual dimorphism in *P. balsamifera*, female flowers are greener than male flowers, which is consistent with the enrichment of the GO terms associated with photosynthesis and chlorophyll production in female-biased genes[86]. We also detected SBGs related to photosynthesis and enriched in female *P. euphratica* catkins in this study. Male *P. euphratica* catkins produce more flowers and accumulate more carbon and nitrogen than female catkins[43,44], which is reflected by the enrichment of starch and sucrose metabolism in genes with male-biased and limited expression identified in this study. In line with our result, a recent study also identified *Potra2n18c32253* (*A. thaliana* synonym is sugar transporter protein 7, *STP7*) was differentially expressed in female and male flower buds in *P. tremula*[87]. The concerted regulation of the transcriptome and methylome may modulate gene expression to meet the greater demands in the energy of male catkins to generate more pollen and increase the probability of successful mating via wind pollination[88]. For instance, a recent study examined the extent of dimorphism in secondary sexual characteristics in *P. trichocarpa* and *P. balsamifera*, but failed to uncover any sexual dimorphic traits for morphology or physiology in vegetative tissues[89]. However, several physiological and metabolic characteristics between males and females appear to differ substantially in their response to chilling or infection in the case of *P. cathayana*[39,40]. Furthermore, according to our observations in the field, male and female *P. euphratica* trees exhibit differences in stress and disease resistance (Fig. 5d). In line with the above traits, we detected extensive sexual dimorphism in gene expression, as we obtained 617 SLGs and 104 SBGs in *P. euphratica* leaves, far greater numbers than the one SBG in *P. balsamifera*[86] and two differentially expressed genes between the two sexes in *P. tremula*[90].

In conclusion, we independently sequenced and de novo assembled the female (XX) and male (XY) *P. euphratica* genomes and identified the X and Y chromosomes. These results lay the foundation for investigating the mechanism of sex determination in *P. euphratica*. We identified the relatively complete SLR-Y and SLR-X regions in *P. euphratica*, and showed its SLR-Y follows the proposed model of evolutionary dynamics for a sex-linked region. The master regulator in the SLR was repeat fragments of *ARR17*, which generate 24-nt siRNAs leading to male-biased methylation of the autosomal gene *ARR17*. Furthermore, the *ARR17* fragment repeats were flanked by palindromic repeats, maintaining the sequence integrity of the SLR, as in other plant species. The SLR also accumulated repetitive sequences that likely contributed to the rapid expansion of MSY. Lastly, we detected two male-specific fusion genes that may be responsible for sexual dimorphism in immune responses between male and female trees. Further functional research on these sex-specific, sex-limited, and sex-biased genes will be important for the understanding of sex determination and sex-specific adaptive evolution.

## Methods

**Plant materials and DNA sequencing**. Male and female *P. euphratica* were collected in Tarim Basin, Xinjiang province, China. Genomic DNA was extracted from leaves (Supplementary Data 1) using the DNAsecure Plant Kit (Tiangen Biotech Co., Ltd., Beijing, China). Genome size, heterozygosity, and repeat contents were surveyed with k-mer analysis to guide the sequencing and assembly of the genome. Based on the estimated heterozygosity of ten female and ten male individuals (Supplementary Data 2), the low rate of 0.86% for the female (XX) genome and the high rate of 1.99% for the male (XY) genome were selected for genome assembly. DNA quality was evaluated with Qubit and Nanodrop to ensure that DNA samples would be appropriate for the construction of Illumina paired-end sequencing libraries (350-bp inserts) and PacBio libraries (20-kb insertion). The resulting male and female libraries were sequenced on a PacBio sequel II platform (Pacific Biosciences, Menlo Park, CA, USA) and an Illumina HiSeq Nova Seq 6000 instrument (Illumina, San Diego, CA, USA). The Hi-C sequencing library was

constructed following published procedures[91] and sequenced using an Illumina Nova Seq 6000 platform (Illumina, San Diego, CA, USA).

**Genome assembly and evaluation**. De novo assembly of the long reads generated from the PacBio SMRT Sequencing was performed using FALCON v1.0[92]. Before assembly, errors in the PacBio reads were corrected with the FALCON pipeline. The corrected reads were then aligned to each other and assembled into genomic contigs using FALCON with the following parameters: length_cutoff_pr = 5000, max_diff = 65, max_cov = 75. The produced primary contigs (p-contigs) were then polished using Quiver by incorporating all SMRT reads. Then, Pilon[93] was used to perform a second round of error correction with short paired-end reads generated from Illumina Nova seq 6000 sequencing. Subsequently, the Purge Haplotigs pipeline[94] was used to remove redundant sequences resulting from heterozygosity in the female individual. Because the male genome had a relatively high degree of heterozygosity, most parts of the genome was assembled as separate primary contigs (haploid assembly), which produced contigs twice the size of the haploid genome (1.03 Gb), rather than a single haplotype-fused contig. Next, the draft diploid assembly and the alignment file of PacBio long-reads mapped to the assembly were loaded into the Purge haplotigs pipeline, resulting in curated primary contigs and reassigned haplotigs.

Hi-C reads were used to improve the quality of assemblies. To avoid reads with artificial bias, we removed the following: (a) reads with ≥10% unidentified nucleotides (N); (b) reads with >10 nt aligned to the adapter, allowing ≤10% mismatches; and (c) reads with >50% bases having phred quality <5. The filtered Hi-C reads were aligned against the contig assemblies with BWA version 0.7.8[95]. Reads were excluded from subsequent analysis if they did not align within 500 bp of a restriction site or did not uniquely map, and the number of Hi-C read pairs linking each pair of scaffolds was tabulated. LACHESIS (http://shendurelab.github.io/LACHESIS/) was used to perform hierarchical agglomerative clustering and assign scaffolds to 19 groups. Juicebox v1.9.8[96] was finally used to order the scaffolds in each group.

BUSCO[47] was used to evaluate the completeness of the assembly and of the annotation by mapping the genome sequence and annotated protein sequences to the database embryophyta_odb10.

**Genome annotation**. Repeat sequences in the genomes were annotated using both ab initio and homology-based search methods. For ab initio predictions, Repeat-Modeler (http://www.repeatmasker.org/RepeatModeler/), RepeatScout (http://www.repeatmasker.org/), and LTR_FINDER (http://tlife.fudan.edu.cn/ltr_finder/) were used to discover TEs and to build a TE library. The resulting TE library and a known repeat library (Repbase V15.02, homolog-based) were then subjected to RepeatMasker (http://www.repeatmasker.org/) for repeat identification. For homology-based predictions, RepeatProteinMask (http://www.repeatmasker.org/) was employed to detect TEs in the genome by comparisons against the TE protein database. A consensus list of TEs was generated from the combination of the two methods. Tandem repeats were identified in the genome using Tandem Repeats Finder (version 4.07b)[97]. All repetitive regions except tandem repeats were soft-masked for gene prediction. Based on the repeat-masked genomes, protein-coding genes were annotated with a combination of RNA-seq mapping, homology-based gene prediction, and ab initio prediction. For RNA-seq prediction, the Illumina RNA-seq data from stems, catkins, leaves, and petioles of the female and male individuals were aligned to the assembled female and male genome using Tophat version 2.0.13[98] to identify exons region and splice positions, respectively. The alignment results were then used as input for Cufflinks version 2.1.1[99] to assemble transcripts to the gene models. In addition, RNA-seq data were assembled by Trinity version 2.1.1[100], creating several pseudo-ESTs. These pseudo-ESTs were also mapped to the assembled genome by BLAT and gene models were predicted using PASA[101]. For protein homolog search, the protein sequences of nine homologous species (*Populus euphratica*, *Populus trichocarpa*, *Populus pruinosa*, *Actinidia chinensis*, *Carica papaya*, *Juglans regia*, *Jatropha curcas*, *Arabidopsis thaliana*, and *Oryza sativa*) were aligned against to the genome using TBLASTN (E-value 1E-05). Genewise version 2.2.0[102] was employed to predict gene models based on these alignments. Ab initio prediction was performed to predict gene structure using five gene prediction programs, including Augustus v3.0.2[103], Genescan v1.0[104], Geneid v1.4[105], GlimmerHMM v3.0.2[106] and SNAP version 2013-02-16[107]. A weighted and non-redundant gene set was generated by EVidenceModeler (EVM, version 1.1.1)[108], which merged all gene models predicted by the above three approaches. Finally, PASA[108] was used to adjust the gene models generated by EVM.

The functional assignments of the *P. euphratica* genes were conducted by the Basic Local Alignment Search Tool (BLAST) against public protein databases, including Swiss-Prot (https://web.expasy.org/docs/swiss-prot_guideline.html), GO, NR, InterPro[109], Pfam[110] and KEGG (https://www.kegg.jp/). ncRNAs were predicted using de novo and homology search methods.

**Identification of the SLR**. DNA sequencing libraries were prepared from bulk genomic DNA from 100 *P. euphratica* female individuals and 96 *P. euphratica* male individuals collected in the wild and sequenced, generating 134-fold and 119-fold coverage, respectively, on an Illumina Nova Seq 6000 instrument. The clean bulked

reads (BSA-M and BSA-F) were aligned and mapped onto the de novo assembled genome from this study by Burrows-Wheeler Aligner software (BWA)[95]. Alignment files were converted to BAM files using SAMtools software[111] with parameter setting as -bS -t. SNP calling and filtering were performed using the Unified Genotype function and the Variant Filtration in GATK software[112]. ANNOVAR software[113] was used to annotate SNPs based on the annotation file for the de novo assembled genome.

To identify the SLR, the ratio of different reads was calculated between the two bulked pools to obtain the SNP-index of each nucleotide position. Subsequently, the difference in SNP-indexes between the two pools was calculated as ΔSNP-index. A sliding window method was used to present the ΔSNP-indexes across the whole genome, with a window size of 1 Mb and a step size of 10 kb. One thousand permutation tests were performed, and the 95% confidence level was selected as the screening threshold.

Moreover, palindromic repeats in SLR-Y were detected based on self-alignment using LASTZ 1.03.66[114] with the following options:–gapped–exact = 100–step = 20. Additionally, small RNA data of female flower buds (BIG BioSample ID: SAMC206760) and male flower buds (BIG BioSample ID: SAMC206761) of *P. euphratica*, downloaded from BIG, were mapped to the SLR-Y and the counterpart region on X (SLR-X) with Bowtie v1.2.2[115]. The mapping results were visualized in the Integrative Genomic Viewer tool[116]. Homologous gene pairs between SLR-Y and SLR-X were identified by performing a reciprocal BLASTP with default parameters.

**Transcriptome sequencing and identification of SLGs/SBGs**. Both leaf and catkin samples from five male and five female individuals of *Populus euphratica* were collected from natural populations in Tarim Basin, Xinjiang province, China, in early spring 2019. Whole male and female catkins were collected when the catkin was fully expanded but had not flowered yet. Young leaves were sampled when they were fully expanded. The tissue samples were frozen in liquid nitrogen and stored at −80 °C until RNA extraction.

Total RNA was extracted from each sample using the RNAprep Pure Plant Kit (Tiangen), and genomic DNA contamination was removed using RNase-Free DNase I (Tiangen). Isolated purified RNA was used for cDNA library construction, with fragment lengths of approximately 250 bp, using the NEBNext Ultra RNA Library Prep Kit for Illumina (New England Biolabs, Ipswich, MA, USA), according to the manufacturer's instructions.

Sequencing was performed on an Illumina Nova Seq 6000 instrument and 150 bp paired-end reads were generated. Raw reads were trimmed by removing adapter sequences, reads with more than 5% of unknown base calls (N), and low-quality bases (base quality less than 5, Q ≤ 5). Clean paired-end reads were aligned to the de novo assembled genomes from this study using HISAT2 v2.0.4[117] with default parameters (Supplementary Data 31). HTSeq v0.6.1[118] was used to count the read numbers mapped to each gene.

Gene expression was quantified using fragments per kilobase of transcript sequence per million base pairs (FPKM). Expressed genes in one tissue were filtered for FPKM >1. In one tissue, genes were classified as exhibiting a sex-limited expression profile if they were expressed in one sex only. Genes with sex-biased expression were expressed in both sexes but differentially expressed in one or the other sex. Normalized read counts for all detected genes were used to identify differentially expressed genes by the negative binomial distribution model implemented in DESeq Software v1.10.1[119]. Differentially expressed genes were identified with a |log$_2$ (fold change)| >1 and adjusted $P$ value <0.05 using the Benjamini and Hochberg correction.

Furthermore, GO enrichment analysis was performed with the GOseq R package v1.24.0[120] with correction for gene length bias. Enrichment $P$ values were estimated with Fisher's exact test.

**Whole-genome bisulfite sequencing and differential sex methylation analysis**. *P. euphratica* samples used for methylation analysis were collected from two environmentally divergent fields and consisted of three males and three females in each environment. Xylem was chosen because it is considered a consistent and homogeneous tissue with little phenotypic plasticity relative to the environment, unlike leaves[77]. Genomic DNA was extracted and fragmented by sonication to 200–300 bp with Covaris S220, followed by end repair and adenylation. Cytosine-methylated barcodes were ligated to sonicated DNA following the manufacturer's instructions. These ligated fragments were treated twice with bisulfite using EZ DNA Methylation-Gold Kit (Zymo Research), before the resulting single-stranded DNA fragments were PCR amplified using KAPA HiFi HotStart Uracil + ReadyMix (2X). The prepared library was sequenced, giving an average depth of 30× on an Illumina Nova seq 6000 instrument.

The raw sequencing reads were trimmed through Trimmomatic v0.36[121] using the parameters: SLIDINGWINDOW: 4:15; LEADING:3, TRAILING:3; ILLUMINACLIP: adapter.fa: 2: 30: 10; MINLEN:36. The trimmed reads for each individual were mapped to the corresponding corrected pseudo-reference genome, which is generated by replacing the FG assembled in this study with filtered SNP variants, using bowtie2[115], and duplicated reads were subsequently removed from the BAM files using SAMtools software[111] with the command: rmdup. Finally, methylation levels were called and extracted using Bismark v0.16.3[122]. Differentially methylated regions (DMRs) between female and male individuals in

each environment were identified using the DSS package[123]. According to the distribution of DMRs across the genome, genes were defined as being related to DMRs when the gene body region (from translation start site to translation termination site) or promoter region (2 kb upstream from the translation start site) overlapped with the DMRs. The genes related to sex-specific DMRs and shared between two environments were used for GO enrichment analysis.

**Statistics and reproducibility**. One thousand permutation test with a screening threshold of 95% confidence level was performed to identify the sex-linked region in the whole genome. Differentially expressed genes and differentially methylated level were taken from distinct biological replicates. $P < 0.05$ after adjustment using the Benjamini and Hochberg correction is considered statistically significant. GO enrichment analysis was performed with the GOseq R package v1.24.0 with correction for gene length bias. Fisher's exact test was performed on enrichment and $P < 0.05$ is considered statically significant.

**Reporting summary**. Further information on research design is available in the Nature Portfolio Reporting Summary linked to this article.

## Data availability

The genome sequencing raw data, BSA sequencing data, RNA-seq data, bisulfite sequencing data, and genome assemblies and annotations for female and male *P. euphratica* in this study have been deposited in the Genome Warehouse in National Genomics Data Center[124,125], Beijing Institute of Genomics, Chinese Academy of Sciences/China National Center for Bioinformation, under BioProject ID PRJCA006811 that is publicly accessible at http://bigd.big.ac.cn/bioproject.

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

## Acknowledgements

This work was financially supported by the National Natural Science Foundation of China (NSFC) to Z.L. (Project No. 31460042), the National Natural Science Foundation of China-Xinjiang Joint Fund to Z.L. (Project No.U1303101), Regional innovation guidance plan project of Xinjiang production and Construction Corps to ZL. (Project No. 2021BB010), Bintuan Science and Technology Program to P.J. (Project No. 2022CB001-10) and Innovative team Building Plan for key areas of Xinjiang Production and Construction Corps to Z.L. (Project No. 2018CB003). We thank Prof. Deborah Charlesworth FRS from the Institute of Evolutionary Biology, University of Edinburgh, for the helpful advice by e-mail communications. We acknowledge Prof. Jianping Liu at Tarim University for their assistance in the research. We thank Prof. Rui Qin and Prof. Hong Liu for providing the learning opportunity at South-Central Minzu University. We are indebted to Zhongshuai Gai, Jianing Wang, Wenrui Qu, and Xiangxiang Chen for their kind help during the preparation of our paper. We also thank Jianhao Sun, Xuefei Guo, Chen Qiu, Xiaoshan Dong, Zijian Wang, Jinlong Zhang, and Yuqi Yang at Tarim University for their assistance in the collection of samples.

## Author contributions

Z.L. and P.J. conceived this project and coordinated research activities; Z.L., D.M., and Z.W. designed experiments; Z.L., J.Z., X.H., and S.Z. collected the samples; S.Z., Z.W., D.M., P.J., Z.J., S.L., and J.X. analyzed the data; S.Z., Z.W., D.M., and Z.L. wrote and revised the manuscript. The authors read and approved the final manuscript.

## Competing interests

The authors declare no competing interests.
