## [Peer Review File · Communications Biology]

Reviewers' comments:

Reviewer #1 (Remarks to the Author):

In this study, the authors presented reference-quality genomes of both sexes of the dioecious tree species *Populus euphratica*. (An improved reference genome of this species was published also in a previous study by Zhang et al. 2020 in *Mol Ecol Res*). *P. euphratica* features an XY sex determination system like most of the other *Populus* species (exceptions known so far: *P. alba* with an ZW-system, Müller et al., *Nature Plants* 2021; *P. adenopoda* and *P. qionghdaoensis* probably with ZW, Kim et al., *Silvae Genetica* 2021). In combination with resequencing data of both sexes they localized and characterized a relatively complete 761-kb sex-linked region (SLR) in the peritelomeric region on chromosome 14 (Y) which is in line with a previous study of Yang et al (*Mol Biol Evol* 2021). As in this previous *P. euphratica* study as well as in studies on other *Populus* species with an XY system (Müller et al., *Nature Plants* 2021; see review Leite Montalvao et al, *Front Plant Sci* 2021), the authors identified partial repeats for the feminizing factor ARR17 (ARABIDOPSIS RESPONSE REGULATOR 17) which was previously identified as the master regulator in a single-factor system of sex determination in *Populus* (Müller et al, *Nature Plants* 2020). The authors elucidated that recombination around the partial ARR17-repeats was potentially suppressed by flanking palindromic arms and the dense accumulation of retrotransposons. The inverted small ARR17-segments S1 and S2 triggered sex determination by generating 24-nt small interfering RNAs that induce male-specific hyper-methylation at the promoter of the autosomal targeted ARR17 as elucidated by the authors. These findings confirm previous results in *P. tremula* (Müller et al. 2020).

The independent chromosome-level genome assemblies for the female (XX) and male (XY) genomes allowed for characterizing the X and Y haplotypes - which differ in gene and repeat content - in more detail resulting in new findings on the evolution of the SDR in this study. Interestingly, an inversion was identified in the SLR-Y overlapping with an old stratum (beside a younger stratum). Two male-specific fusion genes encoding proteins with NB-ARC domains were identified at the breakpoint region of this inversion. These genes were discussed to be responsible for the observed sexual dimorphism in immune responses beside other genes with sex-linked differential expression as identified by RNA-Seq in this study.

1) Please provide a comparison of your SLR with the SDR identified for *P. euphratica* by Yang et al 2021.

2) Please provide at least a short comparison of your reference genomes with the already existing reference genome assembly of Zhang et al 2020.

3) Introduction:

Please consider recent reviews on sex determination in plants (Renner and Müller, *Nature Plants* 2021; Leite Montalvao, *Frontiers in Plant Science* 2021). Please describe the specific features of plant sex chromosomes in more detail where "models of increasing recombination suppression with age do not apply" (Renner and Müller, *Nature Plants* 2021).

4) Please refer to the recent differentiation of single-factor and two-factor systems in plant sex determination (see reviews above).

5) Sentence in lines 65 to 66: I suggest to write: "Member species of the genus *Populus* (Salicaceae family) are exclusively dioecious with various...."

6) I suggest to include - after this sentence - one sentence where you mention that species with XY and ZW systems have been described in *Populus* (review: Leite Montalvao et al. 2021; ZW in *P. alba*; Müller et al 2021 and Yang et al 2020; *P. adenopoda* and *P. qionghdaoensis* probably with ZW, Kim et al., *Silvae Genetica* 2021).

7) Line 67: I suggest to remove "in which"

8) Line 76: Please replace "Similarly" e.g. by "In contrast", because the study in *P. deltoides* describes a two-factor system in contrast to many other studies in *Populus* which provided evidences for a single-factor system of sex determination.

9) Line 82: Please replace "independently" by "independent"

- 10) Line 86: Please replace "Males" by "Male"
- 11) Sentence starting in line 98: "For example.." Please provide a reference for this statement.
- 12) Sentence starting in line 123: "Besides SLR,..." I suggest to write "Outside the SLR, we also identified other....that may contribute to the molecular basis of sexual dimorphism." (only if you mean it)
- 13) Sentence starting in line 262 "Loss-of-function mutation..." I suggest to shift it to the Introduction.
- 14) Figure 3: Please present the small and large fragments of ARR17 in more detail including information on their respective direction.
- 15) Legend to Figure 3: The description of c and d has to be exchanged.
- 16) Line 281 in the legend: Please replace "homolog pairs" by "homolog gene pairs"
- 17) Figure 4a: difficult to understand (would be more clear if you provide a better display of the ARR17 fragments (see my comment 14)
- 18) Please edit the title of Figure 4. What do you mean with "role"?
- 19) Sub-header in line 582: This statement is not correct and has to be edited. The statement is only true for all analysed *Populus* species with an XY system, where RdDM is expected to be the mechanism (a direct proof for RdDM is still missing). In case of *Populus alba* (ZW system), female-specific copies of the entire ARR17 gene on the W-chromosome are the sex-determining genetic factor (see above mentioned review Leite-Montalvao et al). In *Populus alba* males, the ARR17 gene is not present at all.
- 20) Lines 551-553: You might compare your TEs with the TEs identified close to the ARR17 fragments in *P. tremula* (Müller et al, Nature Plants 2020; Extended data fig. 1c). Did you also check for TEs close to the autosomal ARR17 in *P. euphratica*?
- 21) Discussion: Please discuss your RNAseq data also in comparison with other RNAseq studies comparing male and female *Populus* individuals (including Leite Montalvao et al., Philosophical Transactions B 2022).
- 22) Lines 717-719: Please describe the genome annotation in more detail. Which tools using which parameters were applied for which kind of annotation strategy?
- 23) Lines 778-779: Please specify if you used log2FC in addition to adjusted p-values for the selection of differentially expressed genes.
- 24) Lines 798-803: Please provide more details. You assembled two reference genomes. Which one did you use for read mapping? How did you remove optical duplicates? Please provide parameter settings for the tools used.
- 25) Line 825, the link you provided: Please ensure that all data related to this manuscript will be available after potential acceptance of this manuscript. Currently the BioProject related entry is: "This BioProject will be available on 2022-10-10"
- 26) Table S21: Please provide IDs of *Populus trichocarpa* orthologs and functional description terms.

Reviewer #2 (Remarks to the Author):

Poplar is always dioecious, for a few poplar species the sex determination region has been identified. No one before (according to these authors) assembled entire male and female genomes. The work is thorough and lays the ground-work for testing particular hypotheses about the actual mechanism involved in *P. euphratica*.

This group assembled and annotated the male and female genomes of *Populus euphratica*. They started with 20 individuals and discovered the extent of heterozygosity. They picked one male and one female for further analysis. They used Illumina short-read and PacBio long-read sequencing data, with scaffolding informed by chromosomal conformation capture (Hi-C). They had around 100 fold coverage for both types of sequencing. They assembled the two genomes into 19 chromosomes and discovered from the BSA analysis which chromosome has the sex determining locus. Both the male and female genome assemblies displayed good synteny with the *P. trichocarpa* genome. To annotate the genome, they used transcriptomics on

leaves, petioles, catkins and stems. I would have also picked some early inflorescence stages to study sex determination, however, I saw that they have access to such data from a reference.

The BSA analysis used 96 male and 100 female individuals. They determined that the SLR was 760 kb long. There is a large inversion in the SLR. There are also male specific regions, one includes a portion of ARR17, which is the master regulator of sex determination for another poplar species. The male specific regions also had high amounts of repetitive DNA.

Analyzing repetitive DNA, they found two peaks in Gypsy elements on either side of the inversion. They suspect these were responsible for driving the inversion. Using substitution data, they estimated the divergence of SLR-X and SLR-Y within *P. euphratica* occurred about 7.46 Mya, later than the inversion, which dated from 16.44 Mya.

Small RNA data generated from another publication showed that 24-nt RNA reads (>75%), specifically map to one of the male specific regions targeting ARR17. Other genes were differentially methylated as well in this region. They also identified two-novel fusion genes at the breakpoints. They have no way of knowing if these are responsible for sexual dimorphism, but experiments could be performed in the future.

Lots of genes show sex-biased gene expression. This section was a bit boring.

Reviewer #3 (Remarks to the Author):

This paper follows recent paper on sex determination in *Populus* and appears to be thorough and well presented. It is of particular interest because of the phylogenetic position occupied by *P. euphratica*. I have two major comments:

(1) The value of this paper is determined by the quality of the data on which it is based. I therefore went to the data source as stated in the manuscript:

"The genome sequencing raw data, BSA sequencing data, RNA-seq data, bisulfite sequencing data and genome assemblies and annotations for female and male *P. euphratica* in this study have been deposited in the Genome Warehouse in National Genomics Data Center Beijing Institute of Genomics, Chinese Academy of Sciences / China National Center for Bioinformation, under BioProject ID PRJCA006811 that is publicly accessible at <https://ngdc.cncb.ac.cn/gwh>"

I checked and this is not correct, the data is NOT publically accessible: the data and assemblies are under embargo for 1 year (release date June 2023). This is unacceptable. As soon as the paper is published the data on which it is based should be released, otherwise the paper is uncheckable by other researchers and therefore outside the realm of science. The editor should insist that the data are released before publication.

(2) The citation of the literature is generally full and complete. However I noticed that two important papers from the Muller group, that should be discussed in the introduction, are absent:

Leite Montalvão, A. P., Kersten, B., Kim, G., Fladung, M., & Müller, N. A. (2022). ARR17 controls dioecy in *Populus* by repressing B-class MADS-box gene expression. *Philosophical Transactions of the Royal Society B*, 377(1850), 20210217.

Kim, G., Montalvão, A. P. L., Kersten, B., Fladung, M., & Müller, N. A. (2021). The genetic basis of sex determination in *Populus* provides molecular markers across the genus and indicates convergent evolution. *Silvae Genetica*, 70(1), 145-155.

These should be included.

Response to reviewers:

We earnestly appreciate for Editors/Reviewers' warm work during this special time of the COVID-19 pandemic. We would like to thank the reviewers for their insightful and constructive reviews and comments on the manuscript, entitled "**Chromosome-scale assemblies of the male and female *Populus euphratica* genomes reveal the molecular basis of sex determination and sexual dimorphism**". **Revisions are highlighted in blue in this version of manuscript.** The point to point responds to the reviewer's comments are listed as follows:

Reviewers' comments:

Reviewer #1 (Remarks to the Author):

In this study, the authors presented reference-quality genomes of both sexes of the dioecious tree species *Populus euphratica*. (An improved reference genome of this species was published also in a previous study by Zhang et al. 2020 in Mol Ecol Res). *P. euphratica* features an XY sex determination system like most of the other *Populus* species (exceptions known so far: *P. alba* with an ZW-system, Müller et al., Nature Plants 2021; *P. adenopoda* and *P. qionghdaoensis* probably with ZW, Kim et al., Silvae Genetica 2021). In combination with resequencing data of both sexes they localized and characterized a relatively complete 761-kb sex-linked region (SLR) in the peritelomeric region on chromosome 14 (Y) which is in line with a previous study of Yang et al (Mol Biol Evol 2021). As in this previous *P. euphratica* study as well as in studies on other *Populus* species with an XY system (Müller et al., Nature Plants 2021; see review Leite Montalvaio et al, Front Plant Sci 2021), the authors identified partial repeats for the feminizing factor *ARR17* (*ARABIDOPSIS RESPONSE REGULATOR 17*) which was previously identified as the master regulator in a single-factor system of sex determination in *Populus* (Müller et al, Nature Plants 2020). The authors elucidated that recombination around the partial *ARR17*-repeats was potentially suppressed by flanking palindromic arms and the dense accumulation of retrotransposons. The inverted small *ARR17*-segments S1 and S2 triggered sex determination by generating 24-nt small interfering RNAs that induce male-specific hyper-methylation at the promoter of the autosomal targeted *ARR17* as elucidated by the authors. These findings confirm previous results in *P. tremula* (Müller et al. 2020).

The independent chromosome-level genome assemblies for the female (XX) and male (XY) genomes allowed for characterizing the X and Y haplotypes - which differ in gene and repeat content - in more detail resulting in new findings on the evolution of the SDR in this study. Interestingly, an inversion was identified in the SLR-Y overlapping with an old stratum (beside a younger stratum). Two male-specific fusion genes encoding proteins with NB-ARC domains were identified at the breakpoint region of this inversion. These genes were discussed to be responsible for the observed sexual dimorphism in immune responses beside other genes with sex-linked differential expression as identified by RNA-Seq in this study.

(1) Please provide a comparison of your SLR with the SDR identified for *P. euphratica* by Yang et al 2021.

Response: Thanks for your patient review. According to your suggestions, we have performed the synteny analysis between our SLR (“SLR-Y”) and reported SDR (denoted as “SLR-MTY”) by Yang et al. ¹. Firstly, our identified SLR-Y with a length of 761 kb is 103 kb longer than that of previous SLR-MTY with a length of 658 kb. Secondly, we performed synteny analysis by aligning our “SLR-Y” to the “SLR-MTY” using nucmer 4.0.0 in MUMmer tool package ². A large fragment inversion was detected between the two SLRs, corresponding to the region from 24,215 kb to 24,453 kb in our “SLR-Y”. In the previous work ¹, the assembled contig from the female was anchor to chromosome by syntenic relationship with the male genome, thus the large fragment inversion was not identified (Supplementary Figure S6). By contrast, both chromosome 14X and chromosome 14Y were de novo assembled in our study. By syntenic analysis and PacBio long reads spanning the two breakpoints of the inverted repeat, we inferred the existence of this large fragment inversion between SLR-Y and SLR-X. Therefore, our compared result indicates that our SLR assembly is more complete than SLR-MTY (Supplementary Figure S6). Also, the sentence of “These results suggested that the peritelomeric SLR-Y identified here is relatively complete” was revised as “Together with the synteny analysis between the SLR-Y and the reported sex-determining region ¹ of *P. euphratica* (Supplementary Figure S6), these results suggested that the peritelomeric SLR-Y identified here is relatively complete.” in the revised manuscript (Lines 256 - 258).

Supplementary Figure S6 Synteny analysis on SLR-Y in this study and SDR identified in Yang et al. ¹ (denoted as “SLR-MTY” and indicated as yellow region) for *P. euphratica*.

References:

- 1 Yang, W. *et al.* A general model to explain repeated turnovers of sex determination in the Salicaceae. *Molecular Biology and Evolution* **38**, 968-980, doi:10.1093/molbev/msaa261 (2020).
- 2 Kurtz, S. *et al.* Versatile and open software for comparing large genomes. *Genome Biol.* **5**, doi:10.1186/gb-2004-5-2-r12 (2004).

(2) Please provide at least a short comparison of your reference genomes with the already existing reference genome assembly of Zhang et al 2020.

Response: Thanks for your patient review and valuable suggestions. We have added a compared statistical analysis about the assembled genomes. Compared with the genome assembly of Zhang et al. ¹ (GWHAAYU00000000.1), contig N50 of our FG is more than twice of their length (2039 kb vs. 900 kb) (Supplementary Table S3). Our male genome (MG) has the approximately equal contig N50 (892 kb vs. 900 kb). These results indicated the FG assembled in this study has the more continuity than the reported genome. Therefore, the description of “The primary assembly FG

consisted of 432 contigs with an N50 of 2,039 kb, which was higher than that (900 kb) of the previously reported genome ¹ (Table 1; Supplementary Table S3).” was added in lines 156 - 158 of the revised manuscript.

Supplementary Table S3. Assembly statistics for the new assembled genomes in this study and the existing genome (Zhang et al. 2020) on *P. euphratica*.

	FG	TM7.1	TM7.2 (MG)	GWHAAAYU00000000.1 (Zhang et al. 2020)
Sequencing platform	Illumina Nova seq 6000 platform + PacBio sequel II platform	Illumina Nova seq 6000 platform + PacBio sequel II platform		Illumina HiSeq X Ten platform + PacBio Sequel platform
Illumina seq data (coverage)	56.61 Gb (100.33×)	59.76 Gb (110.14×)		25.47 Gb (42.17×)
PacBio seq data (coverage)	50.85 Gb (90.11×)	53.05 Gb (97.77×)		40.24 Gb (66×)
Contig N50 (bp)	2,039,164	892,045		900,000
Scaffold N50(bp)	23,893,804	24,550,255	22,951,477	28,590,778
BUSCO	92.50%	95.90%		91%

Zhang, Z. et al. Improved genome assembly provides new insights into genome evolution in a desert poplar (*Populus euphratica*). *Molecular Ecology Resources* 20, 781-794, doi:10.1111/1755-0998.13142 (2020).

References:

- 1 Zhang, Z. *et al.* Improved genome assembly provides new insights into genome evolution in a desert poplar (*Populus euphratica*). *Molecular Ecology Resources* **20**, 781-794, doi:10.1111/1755-0998.13142 (2020).

(3) Introduction:

Please consider recent reviews on sex determination in plants (Renner and Müller, Nature Plants 2021; Leite Montalvao, Frontiers in Plant Science 2021). Please describe the specific features of plant sex chromosomes in more detail where “models of increasing recombination suppression with age do not apply” (Renner and Müller, Nature Plants 2021).

Response: Thanks for your comments. We have added the detailed description and cited these two references in the introduction that “Additionally, there was proposed a unified sex determination model: similar genes and pathways may determine the sex of several dioecious species, regardless of one-factor or two-factor model ¹” in lines 57 - 59 and “On the other hand, if extended regions that suppressing recombination of sex-determining gene (or genes) do not evolve first, the size of sex-determining regions may decoupled with their age, and sex chromosomes may remain homomorphic ²” in lines 70 - 73 of our revised manuscript.

References:

- 1 Leite Montalvão, A. P., Kersten, B., Fladung, M. & Müller, N. A. The Diversity and Dynamics of Sex Determination in Dioecious Plants. *Front. Plant Sci.* **11**, doi:10.3389/fpls.2020.580488 (2021).
- 2 Renner, S. S. & Müller, N. A. Plant sex chromosomes defy evolutionary models of expanding recombination suppression and genetic degeneration. *Nat. Plants* **7**, 392-402, doi:10.1038/s41477-021-00884-3 (2021).

(4) Please refer to the recent differentiation of single-factor and two-factor systems in plant sex determination (see reviews above).

Response: Thank you for your comment. Two-factor and one-factor models were cited in the introduction of the revised manuscript (Lines 50 - 62).

Please see details in the following:

“The canonical two-factor model for the emergency of dioecy assumes that two sex-determining genes will become linked on one chromosome, one affecting female function and one male function ^{1,2}, which is supported by experimental data of kiwifruit (*Actinidia deliciosa*) ³ and asparagus (*Asparagus officinalis*) ⁴. However, one-factor model proposing a single master regulator of sex switch gene *ARR17* is also raised and verified by CRISPR-Cas9-induced mutation in *Populus tremula* ⁵. Additionally, there was proposed a unified sex determination model: similar genes and pathways may determine the sex of several dioecious species, regardless of one-factor or two-factor model ⁶.

After the emergence of sex determination gene (or genes), a pivotal event in sex chromosome evolution is the suppression of recombination flanking this gene (or these genes)^{1,7-9}.”

References:

- 1 Henry, I. M., Akagi, T., Tao, R. & Comai, L. One Hundred Ways to Invent the Sexes: Theoretical and Observed Paths to Dioecy in Plants. *Annu. Rev. Plant Biol.* **69**, 553-575, doi:10.1146/annurev-arplant-042817-040615 (2018).
- 2 Renner, S. S. & Müller, N. A. Plant sex chromosomes defy evolutionary models of expanding recombination suppression and genetic degeneration. *Nat. Plants* **7**, 392-402, doi:10.1038/s41477-021-00884-3 (2021).
- 3 Akagi, T. *et al.* Two Y-chromosome-encoded genes determine sex in kiwifruit. *Nat. Plants* **5**, 801-809, doi:10.1038/s41477-019-0489-6 (2019).
- 4 Harkess, A. *et al.* Sex Determination by Two Y-Linked Genes in Garden Asparagus. *The Plant*

- Cell* **32**, 1790, doi:10.1105/tpc.19.00859 (2020).
- 5 Müller, N. A. *et al.* A single gene underlies the dynamic evolution of poplar sex determination. *Nat. Plants* **6**, 630-637, doi:10.1038/s41477-020-0672-9 (2020).
- 6 Leite Montalvão, A. P., Kersten, B., Fladung, M. & Müller, N. A. The Diversity and Dynamics of Sex Determination in Dioecious Plants. *Front. Plant Sci.* **11**, doi:10.3389/fpls.2020.580488 (2021).
- 7 Charlesworth, D. Young sex chromosomes in plants and animals. *New Phytol.* **224**, 1095-1107, doi:10.1111/nph.16002 (2019).
- 8 Charlesworth, B. & Charlesworth, D. A Model for the Evolution of Dioecy and Gynodioecy. *The American Naturalist* **112**, 975-997, doi:10.1086/283342 (1978).
- 9 Ming, R., Bendahmane, A. & Renner, S. S. Sex Chromosomes in Land Plants. *Annu. Rev. Plant Biol.* **62**, 485-514, doi:10.1146/annurev-arplant-042110-103914 (2011).

(5) Sentence in lines 65 to 66: I suggest to write: “Member species of the genus *Populus* (Salicaceae family) are exclusively dioecious with various....”

Response: Thanks for your suggestion. We have revised the sentence from “The *Populus*, belongs to Salicaceae family, is exclusively dioecy with various sexual systems, sex chromosomes, and sex-determination regions (SDRs)¹⁻⁵.” as “Member species of the genus *Populus* (Salicaceae family) are exclusively dioecy with various sexual systems, sex chromosomes, and sex-determination regions (SDRs)¹⁻⁵.” in this version of manuscript (Lines 74 - 76).

References:

- 1 Yang, W. *et al.* A general model to explain repeated turnovers of sex determination in the Salicaceae. *Molecular Biology and Evolution* **38**, 968-980, doi:10.1093/molbev/msaa261 (2020).
- 2 Xue, L. *et al.* Evidences for a role of two Y-specific genes in sex determination in *Populus deltoides*. *Nat. Commun.* **11**, 5893, doi:10.1038/s41467-020-19559-2 (2020).
- 3 Müller, N. A. *et al.* A single gene underlies the dynamic evolution of poplar sex determination. *Nat. Plants* **6**, 630-637, doi:10.1038/s41477-020-0672-9 (2020).
- 4 Kim, G., Leite Montalvao, A. P., Kersten, B., Fladung, M. & Müller, N. The genetic basis of sex determination in *Populus* provides molecular markers across the genus and indicates convergent evolution. *Silvae Genet.* **70**, 145-155, doi:10.2478/sg-2021-0012 (2021).
- 5 Leite Montalvão, A. P., Kersten, B., Fladung, M. & Müller, N. A. The Diversity and Dynamics of Sex Determination in Dioecious Plants. *Front. Plant Sci.* **11**, doi:10.3389/fpls.2020.580488 (2021).

(6) I suggest to include - after this sentence - one sentence where you mention

that species with XY and ZW systems have been described in *Populus* (review: Leite Montalvao et al. 2021; ZW in *P. alba*; Müller et al 2021 and Yang et al 2020; *P. adenopoda* and *P. qiongdaoensis* probably with ZW, Kim et al., *Silvae Genetica* 2021).

Response: Thanks for your suggestion. We have added the description as “Among them, *P. alba* (probably also *P. adenopoda* and *P. qiongdaoensis*) has a ZW system¹⁻³, while the rest species in this genus bear a XY system¹⁻⁵.” in the revised manuscript (Lines 76 - 77).

References:

- 1 Yang, W. *et al.* A general model to explain repeated turnovers of sex determination in the Salicaceae. *Molecular Biology and Evolution* **38**, 968-980, doi:10.1093/molbev/msaa261 (2020).
- 2 Kim, G., Leite Montalvao, A. P., Kersten, B., Fladung, M. & Müller, N. The genetic basis of sex determination in *Populus* provides molecular markers across the genus and indicates convergent evolution. *Silvae Genet.* **70**, 145-155, doi:10.2478/sg-2021-0012 (2021).
- 3 Müller, N. A. *et al.* A single gene underlies the dynamic evolution of poplar sex determination. *Nat. Plants* **6**, 630-637, doi:10.1038/s41477-020-0672-9 (2020).
- 4 Xue, L. *et al.* Evidences for a role of two Y-specific genes in sex determination in *Populus deltoides*. *Nat. Commun.* **11**, 5893, doi:10.1038/s41467-020-19559-2 (2020).
- 5 Leite Montalvão, A. P., Kersten, B., Fladung, M. & Müller, N. A. The Diversity and Dynamics of Sex Determination in Dioecious Plants. *Front. Plant Sci.* **11**, doi:10.3389/fpls.2020.580488 (2021).

(7) Line 67: I suggest to remove “in which”

Response: Thanks for your suggestion. We have removed “in which” in the revised manuscript (Line 78).

(8) Line 76: Please replace “Similarly” e.g. by “In contrast”, because the study in *P. deltoides* describes a two-factor system in contrast to many other studies in *Populus* which provided evidences for a single-factor system of sex determination.

Response: OK. We have replaced “Similarly” as “In contrast” in the revised manuscript (Line 87).

(9) Line 82: Please replace “independently” by “independent”

Response: OK. We have replaced the word “independently” by “independent” in the revised manuscript (Line 95).

(10) Line 86: Please replace “Males” by “Male”

Response: OK. We have replaced the word “Males” by “Male” in the revised manuscript (Line 105).

(11) Sentence starting in line 98: “For example..” Please provide a reference for this statement.

Response: OK. We have added the reference in the sentence “For example, male *P. tremuloides* trees have a higher net photosynthetic rate than female trees at elevated CO₂ concentrations” as “For example, male *P. tremuloides* trees have a higher net photosynthetic rate than female trees at elevated CO₂ concentrations¹” in the revised manuscript. (Line 111).

Reference:

- 1 Wang, X. & Curtis, P. S. Gender-specific responses of *Populus tremuloides* to atmospheric CO₂ enrichment. *New Phytol.* **150**, 675-684, doi:<https://doi.org/10.1046/j.1469-8137.2001.00138.x> (2001).

(12) Sentence starting in line 123: “Besides SLR,...” I suggest to write “Outside the SLR, we also identified other....that may contribute to the molecular basis of sexual dimorphism.” (only if you mean it)

Response: Thanks for your suggestion. According to your suggestion, the sentence “Besides SLR, we identified other differentially expressed and differentially methylated genes between female and male *P. euphratica* plants that may participate in the molecular basis of sexual dimorphism” was revised as “Outside the SLR, we also identified other differentially expressed and differentially methylated genes between female and male *P. euphratica* plants that may contribute to the molecular basis of sexual dimorphism” in the revised manuscript (Lines 134 - 137).

(13) Sentence starting in line 262 “Loss-of-function mutation...” I suggest to shift it to the Introduction.

Response: Thanks for your suggestion. As you suggested, the sentence “Loss-of-

function mutation of the target feminizing factor *ARR17* in *P. tremula* leads to the transition from female trees to functional male trees¹” was moved from the Results to the Introduction and adjusted as “However, one-factor model proposing a single master regulator of sex switch gene *ARR17* is also raised and verified by CRISPR-Cas9-induced mutation in *Populus tremula*¹” in the revised manuscript (Lines 54 - 56).

Reference:

- 1 Müller, N. A. *et al.* A single gene underlies the dynamic evolution of poplar sex determination. *Nat. Plants* **6**, 630-637, doi:10.1038/s41477-020-0672-9 (2020).

(14) Figure 3: Please present the small and large fragments of ARR17 in more detail including information on their respective direction.

Response: Thanks for your suggestion. We have presented the detailed information in the revised Figure 3b.

Please see the details in the following:

Fig. 3b Structural variation at SLR-Y.

(15) Legend to Figure 3: The description of c and d has to be exchanged.

Response: OK. As you suggested, the legend of Figure 3c and Figure 3d was exchanged in the revised manuscript (Lines 296 - 297).

(16) Line 281 in the legend: Please replace “homolog pairs” by “homolog gene pairs”

Response: OK. As you suggested, “Types of repeat and repeat density in SLR-Y, and Ks of homolog pairs between SLR-Y and SLR-X” was revised as “Repeat types and repeat density in SLR-Y, and Ks of homolog gene pairs between SLR-Y and SLR-X” in the revised manuscript (Lines 296 - 297).

(17) Figure 4a: difficult to understand (would be more clear if you provide a

better display of the ARR17 fragments (see my comment 14)

Response: Thanks for your comment. We have revised the Figure 4a to present more details on the ARR17 fragments in the revised manuscript.

Please see details in the following:

Fig. 4a *ARR17* on chromosome 19 is partially duplicated in MSY-III of chromosome 14 (Y).

(18) Please edit the title of Figure 4. What do you mean with “role”?

Response: Thanks for your suggestion. We wanted to summarize the effect of methylation in *P. euphratica*. As suggested by the reviewer, the title of Figure 4 was revised from “Sex determination and role in *P. euphratica*” to “Differential methylation levels of *ARR17* in male and female contributing to the sex determination in *P. euphratica*” in the revised manuscript (Lines 322 - 323).

(19) Sub-header in line 582: This statement is not correct and has to be edited. The statement is only true for all analysed *Populus* species with an XY system, where RdDM is expected to be the mechanism (a direct proof for RdDM is still missing). In case of *Populus alba* (ZW system), female-specific copies of the entire *ARR17* gene on the W-chromosome are the sex-determining genetic factor (see above mentioned review Leite-Montalvao et al). In *Populus alba* males, the *ARR17* gene is not present at all.

Response: Thanks for your suggestion. According to your suggestion, the sub-header

“RdDM is a common mechanism for sex determination in the *Populus* genus” was revised as “Mechanism of sex determination in *P. euphratica*.”, and the sentence “Dioecy evolved independently in different species of the genus *Populus*¹, although they all shared a common mechanism of sex determination” was revised as “Dioecy evolved independently in different species of the genus *Populus*¹, although several species (*P. euphratica*², *P. deltoides*¹, *P. tremula*³, *P. tremuloides*³, *P. trichocarpa*³, and *P. balsamifera*^{3,4}) seems to share a common mechanism of sex determination” in the revised manuscript (Lines 605 - 608).

References:

- 1 Xue, L. *et al.* Evidences for a role of two Y-specific genes in sex determination in *Populus deltoides*. *Nat. Commun.* **11**, 5893, doi:10.1038/s41467-020-19559-2 (2020).
- 2 Yang, W. *et al.* A general model to explain repeated turnovers of sex determination in the Salicaceae. *Molecular Biology and Evolution* **38**, 968-980, doi:10.1093/molbev/msaa261 (2020).
- 3 Müller, N. A. *et al.* A single gene underlies the dynamic evolution of poplar sex determination. *Nat. Plants* **6**, 630-637, doi:10.1038/s41477-020-0672-9 (2020).
- 4 Brautigam, K. *et al.* Sexual epigenetics: gender-specific methylation of a gene in the sex determining region of *Populus balsamifera*. *Scientific Reports* **7**, 45388, doi:10.1038/srep45388 (2017).

(20) Lines 551-553: You might compare your TEs with the TEs identified close to the ARR17 fragments in *P. tremula* (Müller et al, Nature Plants 2020; Extended data fig. 1c). Did you also check for TEs close to the autosomal *ARR17* in *P. euphratica*?

Response: Thanks for your suggestion. In contrast to the high density of Gypsy TEs around the ARR17 fragments in *P. euphratica*, no Gypsy was found around the ARR17 fragments in *P. tremula* (Extended Data Fig.1 in Müller et al. ¹), but two inverted Gypsy TEs were found in the region close to autosomal *ARR17* gene (19.783 Mb – 19.785 Mb) in *P. euphratica* (Figure S1). These results indicated that Gypsy possibly contribute to the formation of the partial repeats of ARR17 in *P. euphratica*.

Please see the details in the following figure:

Figure S1 Repeats identified close to ARR17 in chr.19 and the segments of ARR17 (L1-2, S1-3) in chr.14 (Y) in *P. euphratica*.

References:

- 1 Müller, N. A. *et al.* A single gene underlies the dynamic evolution of poplar sex determination. *Nat. Plants* **6**, 630-637, doi:10.1038/s41477-020-0672-9 (2020).

(21) Discussion: Please discuss your RNAseq data also in comparison with other RNAseq studies comparing male and female *Populus* individuals (including Leite Montalvao et al., Philosophical Transactions B 2022).

Response: Thanks for your suggestion. The studies in *P. tremula*¹ support our results that the sex dimorphism on starch and sucrose metabolism in *P. euphratica*. we have discussed these results as “In line with our result, a recent study also identified *Potra2n18c32253* (*A. thaliana* synonym is sugar transporter protein 7, *STP7*) was differentially expressed in female and male flower buds in *P. tremula*¹.” in the revised manuscript (Lines 656 - 658).

References:

- 1 Leite Montalvão, A. P., Kersten, B., Kim, G., Fladung, M. & Müller, N. A. ARR17 controls dioecy in *Populus* by repressing B-class MADS-box gene expression. *Philosophical Transactions of the Royal Society B: Biological Sciences* **377**, 20210217, doi:doi:10.1098/rstb.2021.0217 (2022).

(22) Lines 717-719: Please describe the genome annotation in more detail. Which tools using which parameters were applied for which kind of annotation strategy?

Response: Thanks for your suggestion. As suggested by the reviewer, tools and parameters used in each annotation strategy were described in the materials and methods of the revised manuscript (Lines 751 - 769).

Please see the details in the following:

“For RNA-seq prediction, the Illumina RNA-seq data from stems, catkins, leaves and petioles of the female and male individuals were aligned to the assembled female and male genome using Tophat version 2.0.13 ¹ to identify exons region and splice positions, respectively. The alignment results were then used as input for Cufflinks version 2.1.1 ² to assemble transcripts to the gene models. In addition, RNA-seq data were assembled by Trinity version 2.1.1 ³, creating several pseudo-ESTs. These pseudo-ESTs were also mapped to the assembled genome by BLAT and gene models were predicted using PASA ⁴. For protein homolog search, the protein sequences of 9 homologous species (*Populus euphratica*, *Populus trichocarpa*, *Populus pruinosa*, *Actinidia chinensis*, *Carica papaya*, *Juglans regia*, *Jatropha curcas*, *Arabidopsis thaliana*, *Oryza sativa*) were aligned against to the genome using TBLASTN (E-value 1E-05). Genewise version 2.2.0 ⁵ was employed to predict gene models based on these alignments. *Ab initio* prediction was performed to predict gene structure using five gene prediction programs including Augustus v3.0.2 ⁶, Genescan v1.0 ⁷, Geneid v1.4 ⁸, GlimmerHMM v3.0.2 ⁹ and SNAP version 2013-02-16 ¹⁰. A weighted and non-redundant gene set was generated by EVidenceModeler (EVM, version 1.1.1) ¹¹, which merged all genes models predicted by the above three approaches. Finally, PASA ¹¹ was used to adjust the gene models generated by EVM.”

References:

- 1 Kim, D. *et al.* TopHat2: accurate alignment of transcriptomes in the presence of insertions, deletions and gene fusions. *Genome Biol.* **14**, R36, doi:10.1186/gb-2013-14-4-r36 (2013).
- 2 Trapnell, C. *et al.* Differential gene and transcript expression analysis of RNA-seq experiments with TopHat and Cufflinks. *Nature Protocols* **7**, 562-578, doi:10.1038/nprot.2012.016 (2012).
- 3 Grabherr, M. G. *et al.* Full-length transcriptome assembly from RNA-Seq data without a reference genome. *Nat. Biotechnol.* **29**, 644-652, doi:10.1038/nbt.1883 (2011).
- 4 Haas, B. J. *et al.* Improving the *Arabidopsis* genome annotation using maximal transcript alignment assemblies. *Nucleic Acids Res.* **31**, 5654-5666, doi:10.1093/nar/gkg770 %J Nucleic Acids Research (2003).
- 5 Birney, E., Clamp, M. & Durbin, R. GeneWise and Genomewise. *Genome Res* **14**, 988-995, doi:10.1101/gr.1865504 (2004).
- 6 Stanke, M., Diekhans, M., Baertsch, R. & Haussler, D. Using native and syntenically mapped cDNA alignments to improve de novo gene finding. *Bioinformatics* **24**, 637-644, doi:10.1093/bioinformatics/btn013 (2008).
- 7 Burge, C. & Karlin, S. Prediction of complete gene structures in human genomic DNA. *J. Mol. Biol.* **268**, 78-94, doi:<https://doi.org/10.1006/jmbi.1997.0951> (1997).

- 8 Alioto, T., Blanco, E., Parra, G. & Guigó, R. Using geneid to Identify Genes. *Curr Protoc Bioinformatics* **64**, e56, doi:<https://doi.org/10.1002/cpbi.56> (2018).
- 9 Majoros, W. H., Pertea, M. & Salzberg, S. L. TigrScan and GlimmerHMM: two open source ab initio eukaryotic gene-finders. *Bioinformatics* **20**, 2878-2879, doi:10.1093/bioinformatics/bth315 %J Bioinformatics (2004).
- 10 Korf, I. Gene finding in novel genomes. *BMC Bioinformatics* **5**, 59, doi:10.1186/1471-2105-5-59 (2004).
- 11 Haas, B. J. *et al.* Automated eukaryotic gene structure annotation using EVidenceModeler and the Program to Assemble Spliced Alignments. *Genome Biol.* **9**, R7, doi:10.1186/gb-2008-9-1-r7 (2008).

(23) Lines 778-779: Please specify if you used log2FC in addition to adjusted p-values for the selection of differentially expressed genes.

Response: Thanks for your valuable suggestion. Yes, $|\log_2\text{FC}| > 1$ and adjusted P value < 0.05 were combined to identify the differentially expressed genes. “The Genes with a P value < 0.05 after adjustment using the Benjamini and Hochberg correction by DESeq were assigned as differentially expressed genes” was emended as “Differentially expressed genes were identified with a $|\log_2(\text{fold change})| > 1$ and adjusted P value < 0.05 using the Benjamini and Hochberg correction” in the revised manuscript (Lines 830 - 832).

(24) Lines 798-803: Please provide more details. You assembled two reference genomes. Which one did you use for read mapping? How did you remove optical duplicates? Please provide parameter settings for the tools used.

Response: Thanks for your suggestion. We have modified “The trimmed reads for all individuals were mapped against the *de novo* assembled genome from this study assembled in this study using bowtie2¹, and optical duplicates were subsequently removed from the BAM files.” as “The trimmed reads for each individual were mapped to corresponding corrected pseudo-reference genome, which is generated by replacing the FG assembled in this study with filtered SNP variants, using bowtie2¹, and duplicated reads were subsequently removed from the BAM files using SAMtools software² with the command: rmdup.” in the revised manuscript (Lines 852 - 856).

References:

- 1 Langmead, B. & Salzberg, S. L. Fast gapped-read alignment with Bowtie 2. *Nature Methods* **9**,

357-U354, doi:10.1038/nmeth.1923 (2012).

- 2 Li, H. *et al.* The Sequence Alignment/Map format and SAMtools. *Bioinformatics* **25**, 2078-2079, doi:10.1093/bioinformatics/btp352 %J Bioinformatics (2009).

(25) Line 825, the link you provided: Please ensure that all data related to this manuscript will be available after potential acceptance of this manuscript. Currently the BioProject related entry is: “This BioProject will be available on 2022-10-10”

Response: Thanks for your comment. All the data will be released as soon as the manuscript was potential acceptance. Reviewer’s link for all the data generated in the manuscript were presented in the Data availability in the revised manuscript.

Accession	Title	Project	Reviewer’s Link
GWHBHNV 00000000	Genome of female Populus euphratica in Tarim Basin	PRJCA 006811	https://ngdc.cnpc.ac.cn/gwh/Assembly/reviewer/GnrXOXRShTvhoWfwijOTYBYjAQmCswecarnijHHFMwAJackZmNelDyFGwpiUNXV
GWHBHOO 00000000	Genome of Populus euphratica from a male individual	PRJCA 006811	https://ngdc.cnpc.ac.cn/gwh/Assembly/reviewer/wFRqDxSLkwwMTlhQDZEpnzbVYwigtkhNvcGqJxaeSDYkifezebcCAgWyKfHYnvp
CRA005312	Epigenetic sexual dimorphism for Populus euphratica	PRJCA 006811	https://ngdc.cnpc.ac.cn/gsa/s/FxRmq5g2
CRA005303	Transcriptomic sexual dimorphism for Populus euphratica	PRJCA 006811	https://ngdc.cnpc.ac.cn/gsa/s/AkqaBwU1
CRA005201	BSA sequencing on female and male Populus euphratica	PRJCA 006811	https://ngdc.cnpc.ac.cn/gsa/s/rC2KuDeN
CRA005200	Genome sequencing on male Populus euphratica	PRJCA 006811	https://ngdc.cnpc.ac.cn/gsa/s/NmDDEj84

(26) Table S21: Please provide IDs of *Populus trichocarpa* orthologs and functional description terms.

Response: Thanks for your valuable suggestion. As suggested by the reviewer, we have added the IDs of *Populus trichocarpa* orthologs and functional description terms in the Supplementary Table S21. The table title was emended from “Supplementary Table S21. Genes, expression analysis and GO terms related to expression in the catkins and leaves of female and male *P. euphratica* individuals.” to “Supplementary Table S22. Genes, expression analysis and GO terms related to expression in the catkins and leaves of female and male *P. euphratica* individuals and each ortholog in *P. trichocarpa* V4 and its functional term”.

Supplementary Table S22. Genes, expression analysis and GO terms related to expression in the catkins and leaves of female and male P. euphratica individuals and each ortholog in P. trichocarpa V4 and its functional term.								
Gene_id	chromosome	Group_fpkms				GO term	Ortholog in P. trichocarpa	Functional term (eggNOG)
		F_C	F_L	M_C	M_L			
PeuTF01G00001	chr01	0.49	0.25	0.99	1.04		Potri.001G000450	
PeuTF01G00002	chr01	139.02	111.29	58.69	85.63		Potri.001G000400	Small subunit of serine palmitoyltransferase-like
PeuTF01G00003	chr01	11.35	0.24	236.80	0.03	GO:0016491,GO:0005507,GO:0055114	Potri.001G000500	L-ascorbate oxidase homolog
PeuTF01G00004	chr01	15.39	0.25	442.24	0.04	GO:0055114,GO:0005507,GO:0016491	Potri.001G000600	L-ascorbate oxidase homolog
PeuTF01G00005	chr01	37.38	33.55	26.78	34.03		Potri.001G000700	Transcription factor Pur-alpha
PeuTF01G00006	chr01	0.58	0.18	0.40	0.22	GO:0007018,GO:0005096,GO:0008017,GO:0003777,GO:0007264,GO:0045298,GO:0005874,GO:0005524,GO:0006606,GO:0007017	Potri.001G000800	Belongs to the TRAFAC class myosin-kinesin ATPase superfamily. Kinesin family
PeuTF01G00007	chr01	113.95	104.38	134.38	124.85	GO:0008484,GO:0008152	Potri.001G000900	
PeuTF01G00008	chr01	0.76	2.57	0.87	3.56		Potri.001G001000	DUF761-associated sequence motif
PeuTF01G00009	chr01	0.00	0.17	0.11	0.14	GO:0009664,GO:0005576,GO:0016020	Potri.001G001100	Belongs to the expansin family
PeuTF01G00010	chr01	38.67	45.86	11.99	43.18		Potri.001G001300	
PeuTF01G00011	chr01	5.06	4.24	10.93	4.81		Potri.001G001400	isoform X1
PeuTF01G00012	chr01	16.31	11.98	21.22	15.48	GO:0046872,GO:0000276,GO:0004842,GO:0005680,GO:0015992,GO:0015986,GO:0008270,GO:0016151,GO:0015078,GO:0016567,GO:0061630,GO:0006464,GO:0005515	Potri.001G001500	E3 ubiquitin-protein ligase
PeuTF01G00013	chr01	48.99	47.61	59.02	49.52	GO:0003824,GO:0006094,GO:0030955,GO:0015976,GO:0006144,GO:0000287,GO:0006096,GO:00009405,GO:0016020	Potri.001G001600	Belongs to the pyruvate kinase family
PeuTF01G00014	chr01	8.66	5.03	8.31	6.21		Potri.001G001700	Glucosidase 2 subunit
PeuTF01G00015	chr01	0.87	3.30	0.27	0.50			
PeuTF01G00016	chr01	90.30	107.45	22.39	113.16		Potri.001G002000	YTH domain family protein
PeuTF01G00017	chr01	1.16	2.34	0.31	2.82	GO:0006508,GO:0004252	Potri.001G002200	Subtilase family
PeuTF01G00018	chr01	3.33	1.58	0.20	1.88	GO:0005524,GO:0006468,GO:0004672	Potri.001G002300	Belongs to the protein kinase superfamily
PeuTF01G00019	chr01	2.42	4.03	0.48	2.43	GO:0003700,GO:0005667,GO:0006355,GO:00435	Potri.001G002400	WRKY DNA-binding domain

Reviewer #2 (Remarks to the Author):

Poplar is always dioecious, for a few poplar species the sex determination region has been identified. No one before (according to these authors) assembled entire male and female genomes. The work is thorough and lays the ground-work for testing particular hypotheses about the actual mechanism involved in *P. euphratica*.

This group assembled and annotated the male and female genomes of *Populus euphratica*. They started with 20 individuals and discovered the extent of heterozygosity. They picked one male and one female for further analysis. They used

Illumina short-read and PacBio long-read sequencing data, with scaffolding informed by chromosomal conformation capture (Hi-C). They had around 100 fold coverage for both types of sequencing. They assembled the two genomes into 19 chromosomes and discovered from the BSA analysis which chromosome has the sex determining locus. Both the male and female genome assemblies displayed good synteny with the *P. trichocarpa* genome. To annotate the genome, they used transcriptomics on leaves, petioles, catkins and stems. I would have also picked some early inflorescence stages to study sex determination, however, I saw that they have access to such data from a reference.

The BSA analysis used 96 male and 100 female individuals. They determined that the SLR was 760 kb long. There is a large inversion in the SLR. There are also male specific regions, one includes a portion of ARR17, which is the master regulator of sex determination for another poplar species. The male specific regions also had high amounts of repetitive DNA.

Analyzing repetitive DNA, they found two peaks in Gypsy elements on either side of the inversion. They suspect these were responsible for driving the inversion. Using substitution data, they estimated the divergence of SLR-X and SLR-Y within *P. euphratica* occurred about 7.46 Mya, later than the inversion, which dated from 16.44 Mya.

Small RNA data generated from another publication showed that 24-nt RNA reads (>75%), specifically map to one of the male specific regions targeting *ARR17*. Other genes were differentially methylated as well in this region. They also identified two-novel fusion genes at the breakpoints. They have no way of knowing if these are responsible for sexual dimorphism, but experiments could be performed in the future.

Lots of genes show sex-biased gene expression. This section was a bit boring.

Response: Thanks for your patient review and comments.

Sex determination and sexual dimorphism are interesting topics for developmental and evolutionary biologists. However, the reports on their molecular mechanisms in *P. euphratica*, even in plants, are very scarce. In this study, we investigated the sex determination and sexual dimorphism in *P. euphratica* based on the multi-omics level of genome, transcriptomes and epigenetics, and proposed the possible regulation on sex

determination and sexual dimorphism for further research in future. Among these, some results in this study were also exemplified by previous reports. For example, we found that the siRNA generated from the Y specific partial repeated ARR17 segments induce de novo DNA methylation on the promoter of *ARR17*. Loss-of-function mutation of the target feminizing factor *ARR17* in *P. tremula* leads to the transition from female trees to functional male trees¹. Different with new gene generated from duplication event that could improve the evolution of sexual dimorphism², the male specific novel fused gene *RGA* (resistance gene analog) may be responsible for the sexual dimorphism in resistance to disease for *P. euphratica* in this study. According to your suggestion, further functional experiments related with these genes in this study will be performed in the following work.

Furthermore, sexual dimorphism in plants is generally considered less morphologically conspicuous than in animals. Differentially expressed genes (DEGs) between the sexes thus may provide a quantitative metric to assess the molecular pathways underlying the phenotypic difference not only in the reproductive tissue, but also secondary sexual dimorphism³⁻⁵. Inspired by the work in *P. balsamifera*³, *Salix paraplesia*⁴, *Salix viminalis*⁶, *Silene latifolia*⁵, we performed the identification of differentially expressed gene to elucidate the sexual dimorphism on *P. euphratica*. The related results indicated sex-biased expression may contribute to secondary sexual dimorphisms in development and stress responses to salt, cold and disease, as well as their balance. Therefore, this section provided more transcriptomic evidence for the further study on sexual dimorphism in *P. euphratica*.

References:

- 1 Müller, N. A. *et al.* A single gene underlies the dynamic evolution of poplar sex determination. *Nat. Plants* **6**, 630-637, doi:10.1038/s41477-020-0672-9 (2020).
- 2 Fairbairn, D. J. in *Encyclopedia of Evolutionary Biology* (ed Richard M. Kliman) 105-113 (Academic Press, 2016).
- 3 Sanderson, B. J., Wang, L., Tiffin, P., Wu, Z. & Olson, M. S. Sex-biased gene expression in flowers, but not leaves, reveals secondary sexual dimorphism in *Populus balsamifera*. *New Phytol.* **221**, 527-539, doi:10.1111/nph.15421 (2019).
- 4 Cai, Z., Yang, C., Liao, J., Song, H. & Zhang, S. Sex-biased genes and metabolites explain morphologically sexual dimorphism and reproductive costs in *Salix paraplesia* catkins. *Hortic. Res.-England* **8**, 125, doi:10.1038/s41438-021-00566-3 (2021).
- 5 Zemp, N. *et al.* Evolution of sex-biased gene expression in a dioecious plant. *Nat Plants* **2**, 16168, doi:10.1038/nplants.2016.168 (2016).

- 6 Darolti, I., Wright, A. E., Pucholt, P., Berlin, S. & Mank, J. E. Slow evolution of sex-biased genes in the reproductive tissue of the dioecious plant *Salix viminalis*. *Mol Ecol* **27**, 694-708, doi:10.1111/mec.14466 (2018).

Reviewer #3 (Remarks to the Author):

This paper follows recent paper on sex determination in *Populus* and appears to be thorough and well presented. It is of particular interest because of the phylogenetic position occupied by *P. euphratica*. I have two major comments:

(1)The value of this paper is determined by the quality of the data on which it is based. I therefore went to the data source as stated in the manuscript: "The genome sequencing raw data, BSA sequencing data, RNA-seq data, bisulfite sequencing data and genome assemblies and annotations for female and male *P. euphratica* in this study have been deposited in the Genome Warehouse in National Genomics Data Center Beijing Institute of Genomics, Chinese Academy of Sciences / China National Center for Bioinformatics, under BioProject ID PRJCA006811 that is publicly accessible at <https://ngdc.cnbc.ac.cn/gwh>" I checked and this is not correct, the data is NOT publically accessible: the data and assemblies are under embargo for 1 year (release date June 2023). This is unacceptable. As soon as the paper is published the data on which it is based should be released, otherwise the paper is uncheckable by other researchers and therefore outside the realm of science. The editor should insist that the data are released before publication.

Response: Thanks for your suggestion. Reviewer's link for all the data generated in the manuscript were presented in the following table (also presented in the Data availability in the revised manuscript):

Accession	Title	Project	Reviewer's Link
GWHBHNV 00000000	Genome of female Populus euphratica in Tarim Basin	PRJCA 006811	https://ngdc.cnbc.ac.cn/gwh/Assembly/reviewer/GnrXOXRShTvhoWfwiJOTYBYjAOmCswcoamjiHHFMwAlackZmNeIDyFGwpiUNXV
GWHBHOO	Genome of Populus euphratica from a male	PRJCA	https://ngdc.cnbc.ac.cn/gwh/Assembly/reviewer/wFRgDxSLkwuwMTlhQDZEpnZbVYwigtkhNyCgGJxaeSDYkifczgbeCAgWyKFHYnvp

00000000	individual	006811	
CRA005312	Epigenetic sexual dimorphism for Populus euphratica	PRJCA 006811	https://ngdc.cnpc.ac.cn/gsa/s/FxRmq5g2
CRA005303	Transcriptomic sexual dimorphism for Populus euphratica	PRJCA 006811	https://ngdc.cnpc.ac.cn/gsa/s/AkqaBwU1
CRA005201	BSA sequencing on female and male Populus euphratica	PRJCA 006811	https://ngdc.cnpc.ac.cn/gsa/s/rC2KuDcN
CRA005200	Genome sequencing on male Populus euphratica	PRJCA 006811	https://ngdc.cnpc.ac.cn/gsa/s/NmDDEj84
CRA005199	Genome sequencing on female Populus euphratica	PRJCA 006811	https://ngdc.cnpc.ac.cn/gsa/s/tamsdOBN

(2)The citation of the literature is generally full and complete. However I noticed that two important papers from the Muller group, that should be discussed in the introduction, are absent:

- a) **Leite Montalvão, A. P., Kersten, B., Kim, G., Fladung, M., & Müller, N. A. (2022). ARR17 controls dioecy in *Populus* by repressing B-class MADS-box gene expression. *Philosophical Transactions of the Royal Society B*, 377(1850), 20210217.**
- b) **Kim, G., Montalvão, A. P. L., Kersten, B., Fladung, M., & Müller, N. A. (2021). The genetic basis of sex determination in provides molecular markers across the genus and indicates convergent evolution. *Silvae Genetica*, 70(1), 145-155.**

These should be included.

Response: Thanks for your valuable suggestions and recommendations. These two works are very important in the field of sex determination of the genus *Populus*,

thus are cited in this version of our manuscript.

The first reference you mentioned (Leite Montalvão et al., 2022, Philosophical Transactions of the Royal Society B) is a continuation of previous work ¹ to explore the downstream pathway of the sex switch ARR17. The gene expressional pattern in female and male flower buds were analyzed and differentially expressed genes (DEGs) were identified in this work, we added the discussion as “In line with our result, a recent study also identified *Potra2n18c32253* (*A. thaliana* synonym is sugar transporter protein 7, *STP7*) was differentially expressed in female and male flower buds in *P. tremula* ²” in lines 656 - 658 of the revised manuscript.

The second reference you mentioned (Kim et al., 2021, Silvae Genetica) gives a comprehensive summary of the XY and ZW sex system in the representative species of the genus *Populus*. And it was cited in the introduction as “Member species of the genus *Populus* (belonging to Salicaceae family) are exclusively dioecy with various sexual systems, sex chromosomes, and sex-determination regions (SDRs) ^{1,3-6}. Among them, *P. alba* (probably also *P. adenopoda* and *P. qionghaoensis*) has a ZW system ^{1,3,5}, while the rest species in this genus bear a XY system ^{1,3-6}” in lines 74 - 77 of the revised manuscript.

References:

- 1 Müller, N. A. *et al.* A single gene underlies the dynamic evolution of poplar sex determination. *Nat. Plants* **6**, 630-637, doi:10.1038/s41477-020-0672-9 (2020).
- 2 Leite Montalvão, A. P., Kersten, B., Kim, G., Fladung, M. & Müller, N. A. ARR17 controls dioecy in *Populus* by repressing B-class MADS-box gene expression. *Philosophical Transactions of the Royal Society B: Biological Sciences* **377**, 20210217, doi:doi:10.1098/rstb.2021.0217 (2022).
- 3 Yang, W. *et al.* A general model to explain repeated turnovers of sex determination in the Salicaceae. *Molecular Biology and Evolution* **38**, 968-980, doi:10.1093/molbev/msaa261 (2020).
- 4 Xue, L. *et al.* Evidences for a role of two Y-specific genes in sex determination in *Populus deltoides*. *Nat. Commun.* **11**, 5893, doi:10.1038/s41467-020-19559-2 (2020).
- 5 Kim, G., Leite Montalvao, A. P., Kersten, B., Fladung, M. & Müller, N. The genetic basis of sex determination in *Populus* provides molecular markers across the genus and indicates convergent evolution. *Silvae Genet.* **70**, 145-155, doi:10.2478/sg-2021-0012 (2021).
- 6 Leite Montalvão, A. P., Kersten, B., Fladung, M. & Müller, N. A. The Diversity and Dynamics of Sex Determination in Dioecious Plants. *Front. Plant Sci.* **11**, doi:10.3389/fpls.2020.580488 (2021).

Best regards,

Zhijun Li, PhD

College of Life Sciences and Technology, Tarim University,

Aral 843300, China

REVIEWERS' COMMENTS:

Reviewer #1 (Remarks to the Author):

Thank you for considering all my comments.

Reviewer #4 (Remarks to the Author):

I was asked by the editors specifically to comment on the rebuttal to Reviewer 3's comments. Reviewer 3 is correct in that the data is not publicly available yet, and is currently embargoed. The authors did provide reviewer links that allow reviewers to see that the data is uploaded, but indeed it is not possible to download the assembly or look at raw data (that I can figure out, at least).

From looking at the publishers website, the data access policy of Communications Biology (from <https://www.nature.com/commsbio/editorial-policies/reporting-standards>) is that "Mandatory deposition of data is required for certain data types; see table below with recommended repositories. Supporting data must be made available to editors and peer reviewers where requested at the time of submission for the purposes of evaluating the manuscript. Any restrictions on sharing must be discussed with the editor at submission who reserves the right to decline the study if these conditions are found to be unduly prohibitive."

So, based on this policy, the data do not seem available to review, which is in breach of the journal's policy, and reviewer 3 is correct. I can appreciate that a thorough peer review of a genome-focused paper would involve a reviewer's desire to touch the assembly and annotation to validate claims in the paper. Currently this is not possible.

Response to reviewers:

We would like to thank the reviewers again for your insightful and constructive comments on the manuscript, entitled “**Chromosome-scale assemblies of the male and female *Populus euphratica* genomes reveal the molecular basis of sex determination and sexual dimorphism**”. Revisions are highlighted in blue in this version of manuscript. The point to point responds to the reviewer’s comments are listed as follows:

Reviewers' comments:**Reviewer #1 (Remarks to the Author):**

Thank you for considering all my comments.

Reviewer #4 (Remarks to the Author):

I was asked by the editors specifically to comment on the rebuttal to Reviewer 3's comments. Reviewer 3 is correct in that the data is not publicly available yet, and is currently embargoed. The authors did provide reviewer links that allow reviewers to see that the data is uploaded, but indeed it is not possible to download the assembly or look at raw data (that I can figure out, at least).

From looking at the publishers website, the data access policy of Communications Biology (from <https://www.nature.com/commsbio/editorial-policies/reporting-standards>) is that "Mandatory deposition of data is required for certain data types; see table below with recommended repositories. Supporting data must be made available to editors and peer reviewers where requested at the time of submission for the purposes of evaluating the manuscript. Any restrictions on sharing must be discussed with the editor at submission who reserves the right to decline the study if these conditions are found to be unduly prohibitive."

So, based on this policy, the data do not seem available to review, which is in breach of the journal's policy, and reviewer 3 is correct. I can appreciate that a thorough peer review of a genome-focused paper would involve a reviewer's desire to touch the

assembly and annotation to validate claims in the paper. Currently this is not possible.

Response: Thanks for your suggestion. All the data generated in the manuscript have been released on Oct 5, 2022 under BioProject ID PRJCA006811 that is publicly accessible at <http://bigd.big.ac.cn/bioproject>.

Accession	Submission	Title	Project	Sample	Status	Operation	Released date
 GWHBHNV00000000	WGS024396	Genome of female Populus euphratica in Tarim Basin	PRJCA006811	SAMC467232	RELEASED		2022-10-05
 GWHBHOO00000000	WGS024443	Genome of Populus euphratica from a male individual	PRJCA006811	SAMC467233	RELEASED		2022-10-05

Accession	Submission ID	Title	Create date	Release date	Status	Operation
CRA005312	subCRA007609	Epigenetic sexual dimorphism for Populus euphratica	2021-11-04	2022-10-05	Checked OK Public	
CRA005303	subCRA007600	Transcriptomic sexual dimorphism for Populus euphratica	2021-11-03	2022-10-05	Checked OK Public	
CRA005201	subCRA007393	BSA sequencing on female and male Populus euphratica	2021-10-17	2022-10-05	Checked OK Public	
CRA005200	subCRA007392	Genome sequencing on male Populus euphratica	2021-10-17	2022-10-05	Checked OK Public	
CRA005199	subCRA007390	Genome sequencing on female Populus euphratica	2021-10-17	2022-10-05	Checked OK Public	

Best regards,

Zhijun Li, PhD

College of Life Sciences and Technology, Tarim University,

Aral 843300, China